# DISCO: Adversarial Defense with Local Implicit Functions

**Chih-Hui Ho**    **Nuno Vasconcelos**
Department of Electrical and Computer Engineering
University of California, San Diego
{chh279, nvasconcelos}@ucsd.edu

## Abstract

The problem of adversarial defenses for image classification, where the goal is to robustify a classifier against adversarial examples, is considered. Inspired by the hypothesis that these examples lie beyond the natural image manifold, a novel *aDversarIal defenSe with local impliCit functiOns* (DISCO) is proposed to remove adversarial perturbations by localized manifold projections. DISCO consumes an adversarial image and a query pixel location and outputs a clean RGB value at the location. It is implemented with an encoder and a local implicit module, where the former produces per-pixel deep features and the latter uses the features in the neighborhood of query pixel for predicting the clean RGB value. Extensive experiments demonstrate that both DISCO and its cascade version outperform prior defenses, regardless of whether the defense is known to the attacker. DISCO is also shown to be data and parameter efficient and to mount defenses that transfers across datasets, classifiers and attacks. Code released. [1]

## 1 Introduction

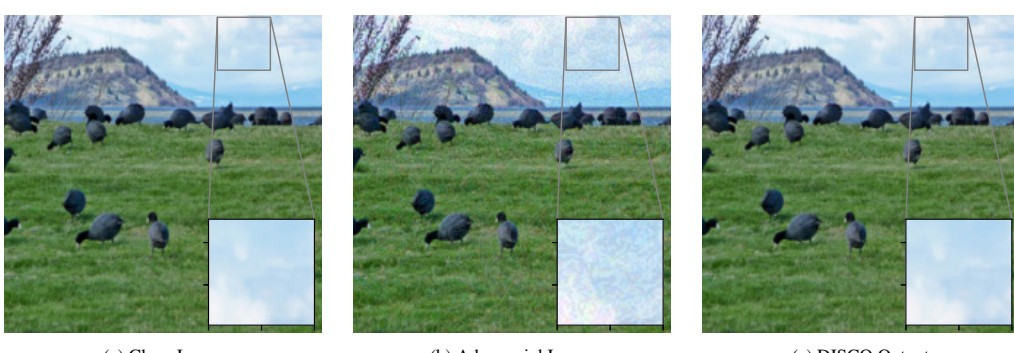

(a) Clean Image           (b) Adversarial Image           (c) DISCO Output

Figure 1: Qualitative performance of DISCO output of a randomly selected ImageNet [23] image.

It has long been hypothesized that vision is only possible because the natural world contains substantial statistical regularities, which are exploited by the vision system to overcome the difficulty of scene understanding [7, 36, 91, 29, 114, 30, 44, 100, 105, 79, 8, 110]. Under this hypothesis, natural images form a low-dimension manifold in image space, denoted as the *image manifold*, to which human vision is highly tuned. While deep neural networks (DNNs) [108, 39, 133, 109, 101] aim to classify natural images with human-like accuracy, they have been shown prone to adversarial

---

[1]Code availabe at `https://github.com/chihhuiho/disco.git`

36th Conference on Neural Information Processing Systems (NeurIPS 2022).

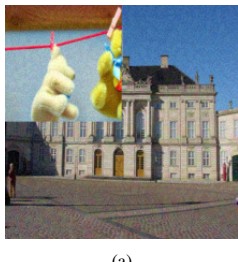 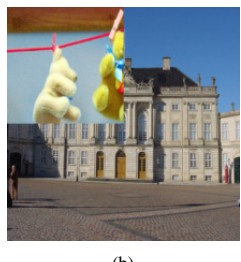 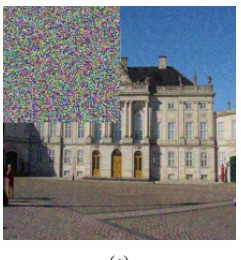 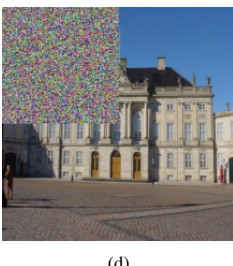

(a)                (b)                (c)                (d)

Figure 2: DISCO is a conditional model of local patch statistics. It performs a local manifold projection per pixel neighborhood, conditional on feature vectors of the adversarial image. This is critical to enable learning from limited data while achieving the model expressiveness needed to precisely control the manifold projection. (a) A mixed image of 2 adversarial images. (b) DISCO output for (a). (c) A mixed image of an adversarial image and noise. (d) DISCO output for (c).

attacks that, although imperceptible to humans, significantly decrease their performance. As shown in Fig. 1 (a) and (b), these attacks typically consist of adding an imperceptible perturbation, which can be generated in various manners [33, 69, 12, 60], to the image. Over the past few years, the introduction of more sophisticated attacks has exposed a significant vulnerability of DNNs to this problem [106, 25, 24, 19, 20, 97]. In fact, it has been shown that adversarial examples crafted with different classifiers and optimization procedures even transfer across networks [22, 90, 125, 46, 83].

A potential justification for the success of adversarial attacks and their transferability is that they create images that lie just barely outside the image manifold [103, 96, 62, 32, 54, 6]. We refer to these images as *barely outliers*. While humans have the ability to project these images into the manifold, probably due to an history of training under adverse conditions, such as environments with low-light or ridden with occlusions, this is not the case for current classifiers. A key factor to the success of the human projection is likely the accurate modeling of the image manifold. Hence, several defenses against adversarial attacks are based on models of natural image statistics. These are usually global image representations [72, 107, 96, 132] or conditional models of image pixel statistics [103, 6, 113, 94, 56]. For example, PixelDefend [103] and HPD [6] project malicious images into the natural image manifold using the PixelCNN model [56], which predicts a pixel value conditioned on the remainder of the image. However, these strategies [103, 6, 72, 107, 96] can be easily defeated by existing attacks. We hypothesize that this is due to the difficulty of learning generative image models, which require *global* image modelling, a highly complex task. It is well known that the synthesis of realistic natural images requires very large model sizes and training datasets [52, 9, 53]. Even with these, it is not clear that the manifold is modeled in enough detail to defend adversarial attacks.

In this work, we argue that, unlike image synthesis, the manifold projection required for adversarial defense is a *conditional* operation: the synthesis of a natural image *given* the perturbed one. Assuming that the attack does not alter the *global* structure of the image (which would likely not be *imperceptible* to humans) it should suffice for this function to be a conditional model of *local* image (i.e. patch) statistics. We argue that this conditional modeling can be implemented with an implicit function [14, 102, 74, 93, 78, 55, 49, 15], where the network learns a conditional representation of the image appearance in the neighborhood of each pixel, given a feature extracted at that pixel. This strategy is denoted *aDversarIal defenSe with local impliCit functiOns* (DISCO). Local implicit models have recently been shown to provide impressive results for 3D modeling [102, 74, 93, 78, 126, 55, 49, 15, 73, 84, 31, 123] and image interpolation [14]. We show that such models can be trained to project barely outliers into the patch manifold, with much smaller parameter and dataset sizes than generative models, while enabling much more precise control of the manifold projection operation. This is illustrated in Fig. 1, which presents an image, its adversarial attack, and the output of the DISCO defense. To train DISCO, a dataset of adversarial-clean pairs is first curated. During training, DISCO inputs an adversarial image and a query pixel location, for which it predicts a new RGB value. This is implemented with a feature encoder and a local implicit function. The former is composed by a set of residual blocks with stacked convolution layers and produces a deep feature per pixel. The latter consumes the query location and the features in a small neighborhood of the query location. The implicit function is learned to minimize the $L_1$ loss between the predicted RGB value and that of the clean image.

The restriction of the manifold modeling to small image neighborhoods is a critical difference with respect to previous defenses based on the modeling of the natural image manifold. Note that, as shown in Fig. 2, DISCO does not project the entire image into the manifold, only each pixel neighborhood. This considerably simplifies the modeling and allows a much more robust defense in a parameter and data efficient manner. This is demonstrated by evaluating the performance of DISCO under both the oblivious and adaptive settings [129, 89, 127]. Under the oblivious setting, the popular RobustBench [17] benchmark is considered, for both $L_\infty$ and $L_2$ attacks with Autoattack [19]. DISCO achieves SOTA robust accuracy (RA) performance, e.g. outperforming the prior art on Cifar10, Cifar100 and ImageNet by 17%, 19% and 19% on $L_\infty$ Autoattack. A comparison to recent test-time defenses [130, 2, 92, 18, 71, 77] also shows that DISCO is a more effective defensive strategy across various datasets and attacks. Furthermore, a study of the defense transferability across datasets, classifiers and attacks shows that the DISCO defense maintains much of its robustness even when deployed in a setting that differs from that used for training by any combination of these three factors. Finally, the importance of the local manifold modeling is illustrated by experiments on ImageNet [23], where it is shown that even when trained with only 0.5% of the dataset DISCO outperforms all prior defenses. Under the adaptive setting, DISCO is evaluated using the BPDA [5] attack, known to circumvent most defenses based on image transformation [103, 96, 107, 127, 27]. While DISCO is more vulnerable under this setting, where the defense is known to the attacker, it still outperforms existing approaches by 46.76%. More importantly, we show that the defense can be substantially strengthened by cascading DISCO stages, which magnifies the gains of DISCO to 57.77%. This again leverages the parameter efficiency of the local modeling of image statistics, which allows the implementation of DISCO cascades with low complexity. The ability to cascade DISCO stages also allows a new type of defense, where the number of DISCO stages is randomized on a per image basis. This introduces some degree of uncertainty about the defense even under the adaptive setting and further improves robustness.

Overall, this work makes four contributions. First, it proposes the use of defenses based purely on the conditional modeling of local image statistics. Second, it introduces a novel defense of this type, DISCO, based on local implicit functions. Third, it leverages the parameter efficiency of the local modeling to propose a cascaded version of DISCO that is shown robust even to adaptive attacks. Finally, DISCO is shown to outperform prior defenses on RobustBench [17] and other 11 attacks, as well as test-time defenses under various experimental settings. Extensive ablations demonstrate that DISCO has unmatched defense transferrability in the literature, across datasets, attacks and classifiers.

## 2 Related Work

**Adversarial Attack and Defense.** We briefly review adversarial attacks and defenses for classification and prior art related to our work. Please refer to [13, 81, 1] for more complete reviews.

*Adversarial Attacks* aim to fool the classifier by generating an imperceptible perturbation (under $L_p$ norm constraint) that is applied to the clean image. Attack methods have evolved from simple addition of sign gradient, as in FGSM [33], to more sophisticated approaches [60, 12, 106, 112, 69, 19, 25, 24, 20]. While most white-box attacks assume access to the classifier gradient, BPDA [5] proposed a gradient approximation attack that can circumvent defenses built on obfuscated gradients. In general, these attacks fall into two settings, oblivious or adaptive, depending on whether the attacker is aware of the defense strategy [129, 89, 127]. DISCO is evaluated under both settings.

*Adversarial Defenses* can be categorized into adversarially trained and transformation based. The former are trained against adversarial examples generated on-the-fly during training [88, 35, 34, 87, 98, 82, 98], allowing the resulting robust classifier to defend against the adversarial examples. While adversarially trained defenses dominate the literature, they are bound together with the classifier. Hence, re-training is required if the classifier changes and the cost of adversarial training increases for larger classifiers. Transformation based defenses [37, 127, 48, 103, 107] instead introduce an additional defense module, which can be applied to many classifiers. This module preprocesses the input image before passing it to the classifier. The proposed preprocessing steps include JPEG compression [21, 27, 68], bit reduction [37, 127, 48], pixel deflection [86] or applications of random transformations [124, 37] and median filters [80]. Beyond pixel space defenses, malicious images can also be reconstructed to better match natural image statistics using autoencoders [72, 107], GANs [96, 4, 132], or other generative models, such as the PixelCNN [94]. The latter is used to

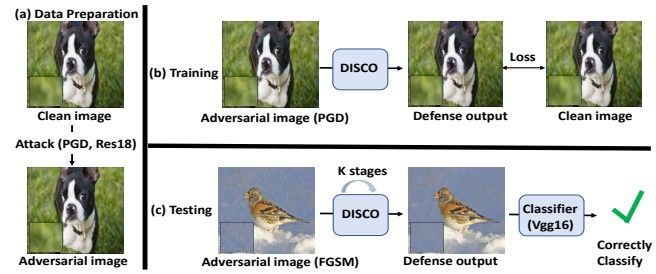

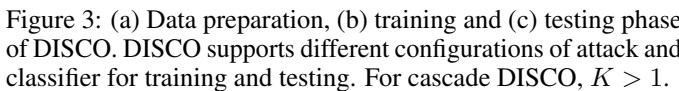

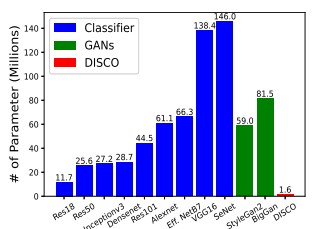

Figure 3: (a) Data preparation, (b) training and (c) testing phase of DISCO. DISCO supports different configurations of attack and classifier for training and testing. For cascade DISCO, $K > 1$.

Figure 4: Number of parameters (Millions) of recent classifiers, GANs and DISCO.

project the malicious image into the natural image manifold by methods like PixelDefend [103] or HPD [6]. These methods can only produce images of fixed size [72, 96, 107, 132] and model pixel likelihoods over the entire image [103, 6]. This is unlike DISCO, which models conditional local statistics and can produce outputs of various size.

The idea of performing adversarial purification before feeding the image into the classifier is central to a recent wave of test-time defenses [130, 2, 71, 77]. [130] addresses the impracticality of previous Monte-Carlo purification models by introducing a Denoising Score-Matching and a random noise injection mechanism. [2] prepends an anti-adversary layer to the classifier, with the goal of maximizing the classifier confidence of the predicted label. [71] reverses the adversarial examples using self-supervised contrastive loss. [77] proposed a diffusion model for adversarial removal. Unlike these prior works, DISCO purifies the adversarial image by modeling the local patch statistics. Such characteristics results in data and parameter efficiency, which have not been demonstrated for [130, 2, 71, 77]. Furthermore, DISCO outperforms all prior works in terms of robust accuracy, under the various settings they proposed.

**Implicit Function.** refers to the use of a neural network to model a continuous function [102]. This has been widely used in applications involving audio [136, 38, 102], 2D images [14, 26] and 3D shapes [102, 74, 93, 78, 126, 55, 49, 15, 73, 63, 84, 31, 123]. In the 3D literature, local implicit functions have become popular models of object shape [78, 93, 84] or complex 3D scenes [74]. This also inspired 2D applications to super-resolution [14], image [51] and video generation [131]. In the adversarial attack literature, implicit functions have recently been proposed to restore adversarial point clouds of 3D shape, through the IF-Defense [123]. To the best of our knowledge, ours is the first paper to propose local implicit functions for 2D adversarial defense.

# 3 Method

In this section, we introduce the architecture of DISCO and its training and testing procedure.

## 3.1 Motivation

Under the hypothesis that natural images lie on a low-dimension image manifold, classification networks can be robustified by learning to project barely outliers (i.e. adversarial images) into the manifold, a process that can be seen as *manifold thickening*. Images in a shell around the manifold are projected into it, leaving a larger margin to images that should be recognized as outliers. While this idea has been studied [32, 54, 66, 47], its success hinges on the ability of classification models to capture the complexities of the image manifold. This is a very hard problem, as evidenced by the difficulty of model inversion algorithms that aim to synthesize images with a classifier [115, 70, 76, 128]. These algorithms fail to produce images comparable to the state of the art in generative modeling, such as GANs [53, 52, 10]. Recently, however, it has been shown that it is possible to synthesize realistic images and 3D shapes with implicit functions, through the use of deep networks [31, 14, 102, 74, 93, 78, 49, 15] that basically memorize images or objects as continuous functions. The assumption behind DISCO is that these implicit functions can capture the local statistics of images or 3D shapes, and can be trained for manifold thickening, that is to learn how to projecting barely outliers into the image manifold.

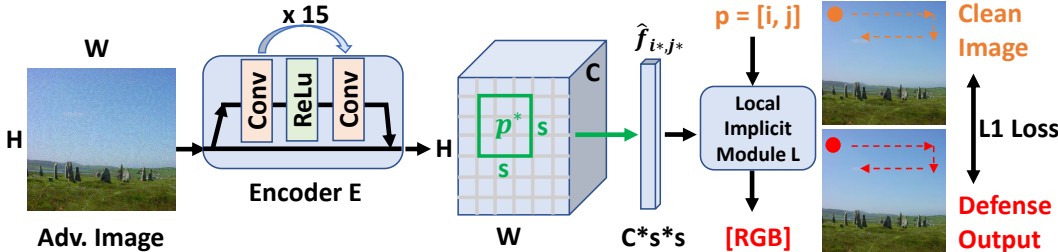

Figure 5: The DISCO architecture includes an encoder and a local implicit module. The network is trained to map Adversarial into Defense images, using an $L_1$ loss to Clean images.

## 3.2 Model Architecture and Training

**Data Preparation** To train DISCO, a dataset $D = \{(x^i_{cln}, x^i_{adv})\}_{i=1}^N$ containing a set of paired clean $x^i_{cln}$ and adversarial $x^i_{adv}$ images is curated. For this, a classifier $P_{trn}$ and an attack procedure $A_{trn}$ are first selected. For each image $x^i_{cln}$, the adversarial image $x^i_{adv}$ is generated by attacking the predictions $P_{trn}(x^i_{cln})$ using $A_{trn}$, as shown in Fig. 3(a).

**Training** As shown in Fig. 3(b), the DISCO defense is trained with pairs of random patches cropped at the same location of the images $x_{cln}$ and $x_{adv}$. For example, random patches of size $48 \times 48$ are sampled from training pairs of the ImageNet dataset [23]. The network is then trained to minimize the L1 loss between the RGB values of the clean $x_{cln}$ and defense output $x_{def}$.

**Architecture** DISCO performs manifold thickening by leveraging the LIIF [15] architecture to purify adversarial patches. It takes an adversarial image $x_{adv} \in \mathbb{R}^{H \times W \times 3}$ and a query pixel location $p = [i, j] \in \mathbb{R}^2$ as input and predicts a clean RGB value $v \in \mathbb{R}^3$ at the query pixel location, ideally identical to the pixel value of the clean image $x_{cln} \in \mathbb{R}^{H \times W \times 3}$ at that location. The defense output $x_{def} \in \mathbb{R}^{H' \times W' \times 3}$ can then be synthesized by predicting a RGB value for each pixel location in a grid of size $H' \times W' \times 3$. Note that it is not a requirement that the size of $x_{def}$ be the same as that of $x_{cln}$. In fact, the size of $x_{def}$ could be changed during inference.

To implement this, DISCO is composed of two modules, illustrated in Fig. 5. The first is an encoder $E$ that extracts the per-pixel feature of an input image $x$. The encoder architecture resembles the design of EDSR [65], originally proposed for super-resolution. It contains a sequence of 15 residual blocks, each composed of a convolution layer, a ReLu layer and a second convolution layer. The encoder output is a feature map $f = E(x) \in \mathbb{R}^{H \times W \times C}$, with $C = 64$ channels. The feature at location $p = [i, j]$ is denoted as $f_{ij}$. The second module is the the local implicit module $L$, which is implemented by a MLP. Given query pixel location $p$, $L$ first finds the nearest pixel value $p^* = [i^*, j^*]$ in the input image $x$ and corresponding feature $f_{i^* j^*}$. $L$ then takes the features in the neighborhood of $p^*$ into consideration to predict the clean RGB value $v$. More specifically, let $\hat{f}_{i^* j^*}$ denote a concatenation of the features whose location is within the kernel size $s$ of $p^*$. The local implicit module $L$ takes the concatenated feature, the relative position $r = p - p^*$ between $p$ and $p^*$, and the pixel shape as input, to predict a RGB value $v$. By default, the kernel size $s$ is set to be 3. Since the network implements a continuous function, based only on neighboring pixels, the original grid size $H \times W$ is not important. The image coordinates can be normalized so that $(i, j) \in [-1, 1]^2$ and the pixel shape is the height and width of the output pixel in the normalized coordinates. This makes DISCO independent of the original image size or resolution.

## 3.3 Inference

For inference, DISCO takes either a clean or an adversarial image as input. Given a specified output size for $x_{def}$, DISCO loops over all the output pixel locations, predicting an RGB value per location. Note that this is not computationally intensive because the encoder feature map $f = E(x)$ is computed once and used to the predict the RGB values of all query pixel locations. Furthermore, while the training pairs are generated with classifier $P_{trn}$ and attack $A_{trn}$, the inference time classifier $P_{tst}$ and attack $A_{tst}$ could be different. In the experimental section we show that DISCO is quite flexible, performing well when (1) $P_{trn}$ and $P_{tst}$ consume images of different input size and (2) the attack, classifier and dataset used for inference are different than those used for training. In fact, DISCO is shown to be more robust to these configuration changes than previous methods.

### 3.4 DISCO Cascades

DISCO is computationally very appealing because it disentangles the training of the defense from that of the classifier. This can be a big practical advantage, since classifier retraining is needed whenever training settings, such as architecture, hyper-parameters, or number of classes, change. Adversarial defenses require retraining on the entire dataset when this is the case, which is particularly expensive for large models (like SENet [41] or EfficientNet [109]) trained on large datasets (like ImageNet [23] and OpenImages [57]). Unsurprisingly, RobustBench [17], one of the largest adversarial learning benchmarks, reports more than 70 baselines for Cifar10, but less than 5 on ImageNet.

DISCO does not have this defense complexity, since it is trained independently of the classifier. Furthermore, because DISCO is a model of local statistics, it is particularly parameter efficient. As shown in Fig. 4, DISCO has a lightweight design with only 1.6M parameters, which is significantly less than most recent classifier [109, 41, 101, 39] and GAN [53, 9] models with good performance for ImageNet-like images. This also leads to a computationally efficient defense. Our experiments show that DISCO can be trained with only 50,000 training pairs. In fact, we show that it can beat the prior SOTA using less than 0.5% of ImageNet as training data (Table 4). One major benefit of this efficiency is that it creates a *large unbalance between the costs of defense and attack.* Consider memory usage, which is dominated by the computation of gradients needed for either the attack or the backpropagation of training. Let $N_d$ and $N_c$ be the number of parameters of the DISCO network and classifier, respectively. The per image memory cost of training the DISCO defense is $O(N_d)$. On the other hand, the attack cost depends on the information available to the attacker. We consider two settings, commonly considered in the literature. In the *oblivious* setting, only the classifier is known to that attacker and the attack has cost $O(N_c)$. In the *adaptive* setting, both the classifier and the DISCO are exposed and backpropagation has memory cost $O(N_c + N_d)$. In experiments, we show that DISCO is quite effective against oblivious attacks. Adaptive attacks are more challenging. However, as shown in Fig 4, it is usually the case that $N_c > N_d$, making the complexity of the attack larger than that of the defense. This is unlike adversarial training, where attack and defense require backpropagation on the same model and thus have the same per-image cost.

This asymmetry between the memory cost of the attack and defense under DISCO can be *magnified* by cascading DISCO networks. If $K$ identical stages of DISCO are cascaded, the defense complexity remains $O(N_d)$ but that of the attack raises to $O(N_c + KN_d)$. Hence the ratio of attack-to-defense cost raises to $O(K + N_c/N_d)$. Interestingly, our experiments (see Section 4.2) show that when $K$ is increased the defense performance of the DISCO cascade increases as well. Hence, DISCO cascades combine high robust accuracy with a large ratio of attack-to-defense cost.

## 4 Experiments

In this section, we discuss experiments performed to evaluate the robustness of DISCO. Results are discussed for both the oblivious and adaptive settings [129, 89, 127] and each result is averaged over 3 trials. $\epsilon_p$ denotes the perturbation magnitude under the $L_p$ norm. All experiments are conducted on a single Nvidia Titan Xp GPU with Intel Xeon CPU E5-2630 using Pytorch [85]. Please see appendix for more training details, quantitative results and visualizations. We adopt the code from LIIF [14] for implementation.

**Training Dataset:** The following training configuration is used unless noted. Three datasets are considered: Cifar10 [58], Cifar100 [59] and Imagenet [23]. For each, 50,000 adversarial-clean training pairs are curated. For Cifar10 and Cifar100, these are the images in the training set, while for ImageNet, 50 images are randomly sampled per class. Following RobustBench [17], the evaluation is then conducted on the test set of each dataset. To create training pairs, PGD [69] ($\epsilon_\infty = 8/255$ with step size is 2/255 and the number of steps 100) is used to attack a ResNet18 and a WideResNet28 on Cifar10/ImageNet and Cifar100, respectively.

**Attack and Benchmark:** DISCO is evaluated on RobustBench [17], which contains more than 100 baselines evaluated using Autoattack [19]. This is an ensemble of four sequential attacks, including the PGD [69] attack with two different optimization losses, the FAB attack [16] and the black-box Square Attack [3]. DISCO is compared to defense baselines under both $L_\infty$ and $L_2$ norms. To study defense generalization, 11 additional attacks are considered, including FGSM [33], BIM [60], BPDA [5] and EotPgd [67]. Note that DISCO is not trained specifically for these attacks.

Table 1: Compare DISCO to the selected baselines on Cifar10 ($\epsilon_\infty = 8/255$).

| Method | SA | RA | Avg. | Classifier |
|---|---|---|---|---|
| No Defense | **94.78** | 0 | 47.39 | WRN28-10 |
| Rebuffi et al. [88] | 92.23 | 66.58 | 79.41 | WRN70-16 |
| Gowal et al. [35] | 88.74 | 66.11 | 77.43 | WRN70-16 |
| Gowal et al. [35] | 87.5 | 63.44 | 75.47 | WRN28-10 |
| Bit Reduction [127] | 92.66 | 1.04 | 46.85 | WRN28-10 |
| Jpeg [27] | 83.9 | 50.73 | 67.32 | WRN28-10 |
| Input Rand. [124] | 94.3 | 8.59 | 51.45 | WRN28-10 |
| AutoEncoder | 76.54 | 67.41 | 71.98 | WRN28-10 |
| STL [107] | 82.22 | 67.92 | 75.07 | WRN28-10 |
| DISCO | 89.26 | **85.56** | **87.41** | WRN28-10 |

Table 2: Compare DISCO to the selected baselines on Cifar10 ($\epsilon_2 = 0.5$).

| Method | SA | RA | Avg. | Classifier |
|---|---|---|---|---|
| No Defense | **94.78** | 0 | 47.39 | WRN28-10 |
| Rebuffi et al. [88] | 95.74 | 82.32 | **89.03** | WRN70-16 |
| Gowal et al. [34] | 94.74 | 80.53 | 87.64 | WRN70-16 |
| Rebuffi et al. [88] | 91.79 | 78.8 | 85.30 | WRN28-10 |
| Bit Reduction [127] | 92.66 | 3.8 | 48.23 | WRN28-10 |
| Jpeg [27] | 83.9 | 69.85 | 76.88 | WRN28-10 |
| Input Rand. [124] | 94.3 | 25.71 | 60.01 | WRN28-10 |
| AutoEncoder | 76.54 | 71.71 | 74.13 | WRN28-10 |
| STL [107] | 82.22 | 74.33 | 78.28 | WRN28-10 |
| DISCO | 89.26 | **88.47** | 88.87 | WRN28-10 |

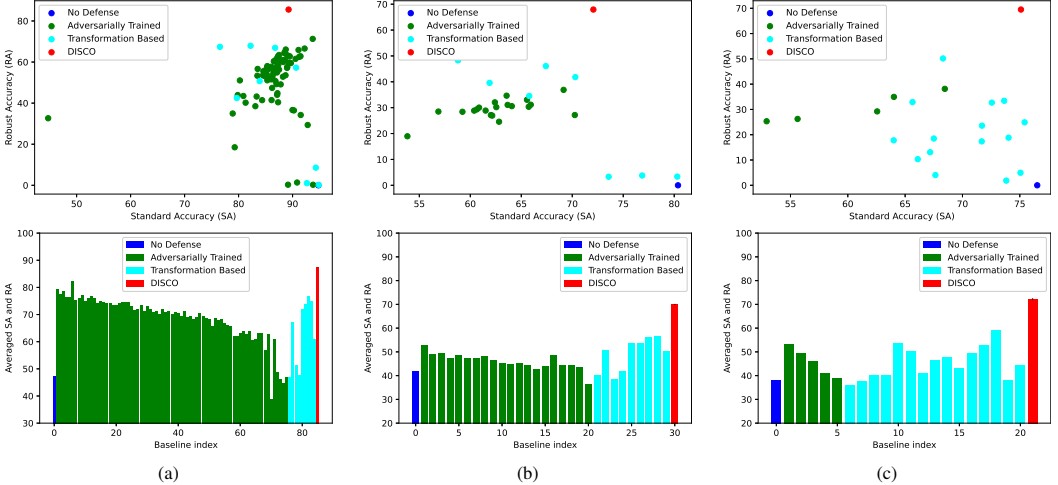

Figure 6: Comparison of DISCO to No Defense, Adversarially Trained, and Transformation based baselines. (a) Cifar10, (b) Cifar100, and (c) ImageNet. Top-row: trade-off between SA and RA. Bottom row: average accuracy of each of the RobustBench baselines and DISCO.

**Metric:** Standard (**SA**) and robust (**RA**) accuracy are considered. The former measures classification accuracy on clean examples, the latter on adversarial. The average of SA and RA is also used.

### 4.1 Oblivious Adversary

**SOTA on RobustBench:** DISCO achieves SOTA performance on RobustBench. Table 1 and 2 compare DISCO to the RobustBench baselines on Cifar10 under $L_\infty$ and $L_2$ Autoattack, respectively. Baselines are categorized into (1) no defense (first block), (2) adversarially trained (second block) and (3) transformation based (third block). The methods presented in each table are those of highest RA performance in each category. The full table is given in the supplemental, together with those of Cifar100 and ImageNet. Note that the RobustBench comparison slightly favours the adversarially trained methods, which use a larger classifier. A detailed comparison to all RobustBench baselines is given in Fig. 6, for the three datasets. The upper row visualizes the trade-off between SA and RA. The bottom row plots the averaged SA/RA across baselines. Blue, green, cyan and red indicate no defense, adversarially trained baselines, transformation based baselines and DISCO, respectively.

These results show that, without a defense, the attack fools the classifier on nearly all examples. Adversarially trained baselines improve RA by training against the adversarial examples. Some of these [88, 35, 34, 50, 87, 43, 120, 104, 122] also leverage additional training data. Transformation based defenses require no modification of the pre-trained classifier and can generalize across attack strategies [96, 107]. While early methods (like Jpeg Compression [27] and Bit Reduction [127]) are not competitive, recent defenses [107] outperform adversarially trained baselines on Cifar100 and ImageNet. DISCO is an even more powerful transformation-based defense, which clearly outperforms the prior SOTA RA by a large margin (17 % on Cifar10, 19 % on Cifar100 and 19 % on ImageNet). In the upper row of Fig. 6, it clearly beats the prior Pareto front for the SA vs. RA trade-off. Table 1

Table 3: Improving ResNet50 baselines on ImageNet.

| Method | SA | RA | Avg. |
|---|---|---|---|
| Hadi et al. [95] | **64.02** | 34.96 | 49.49 |
| w/ DISCO | 63.66 | **50.6** | **57.13** |
| Engstrom et al. [28] | **62.56** | 29.22 | 45.89 |
| w/ DISCO | 62.48 | **49.44** | **55.96** |
| Wong et al. [118] | **55.62** | 26.24 | 40.93 |
| w/ DISCO | 54.52 | **40.68** | **47.6** |

Table 4: Ablation of sampled classes.

| Cls. # | Dataset size | SA | RA | Avg. |
|---|---|---|---|---|
| 100 | 5000 | **72.78** | 59.84 | 66.31 |
| 500 | 25000 | 71.88 | 67.84 | 69.86 |
| 1000 | 50000 | 72.64 | **68.2** | **70.42** |

Table 5: Defense Transfer of $L_\infty$ trained defenses to $L_2$ attacks on Cifar10. Top block: adversarially trained, Bottom block: transformation based.

| Method | Classifier | Clean | FGSM | BIM | CW | DeepFool |
|---|---|---|---|---|---|---|
| Adv. FGSM | ResNet | 91 | **91** | **91** | 7 | 0 |
| Adv. BIM | ResNet | 87 | 52 | 32 | **42** | **48** |
| PixelDefend [103] | VGG | 85 | 46 | 46 | 78 | 80 |
| PixelDefend [103] | ResNet | 82 | 62 | 61 | 79 | 76 |
| PixelDefend (Adv.) [103] | ResNet | 88 | 68 | 69 | 84 | 85 |
| Feature Squeezing [127] | ResNet | 84 | 20 | 0 | 78 | N/A |
| EGC-FL [132] | ResNet | 91.65 | 88.51 | 88.75 | **90.03** | N/A |
| STL [107] | VGG16 | 90.11 | 87.15 | 88.03 | 89.04 | 88.9 |
| DISCO | WRN28 | 89.26 | **89.53** | **89.58** | 89.3 | **89.58** |

and Table 2 also show that previous transformation based methods tend to perform better for $L_2$ than $L_\infty$ Autoattack. DISCO is more robust, achieving similar RA for $L_2$ and $L_\infty$ Autoattacks.

**Improving SOTA Methods:** While DISCO outperforms the SOTA methods on RobustBench, it can also be combined with the latter. Table 3 shows that adding DISCO improves the performance of top three ResNet50 robust baselines for ImageNet [95, 28, 118] by 16.77 (for RA) and 8.12 (for averaged SA/RA) on average. This demonstrates the plug-and-play ability of DISCO.

**Comparison to Test-Time Defenses** First, we compare DISCO to four recent test-time defenses. Following the setup of [130], DISCO is evaluated on Cifar10 using a WRN28-10 network under the PGD40 attack ($\epsilon = 8/255$). While [130] reported an RA of 80.24 for the default setting, DISCO achieves 80.80, even though it is not optimized for this experiment and has much fewer parameters (1.6M vs 29.7M). Second, a comparison to [2, 71] under Autoattack, shows that [2] achieves RAs of 79.21/40.68 and [71] of 67.79/33.16 on the Cifar10/Cifar100 datasets. These numbers are much lower than those reported for DISCO (85.56/67.93) on Table 1 & Appendix Table C. Third, under the APgd [20] attack, [2] achieves 80.65/47.63 RA on Cifar10/Cifar100 dataset, while DISCO achieves 85.79/77.33 (Appendix Table E & Table 3). This shows that DISCO clearly outperforms [2] on two different attacks and datasets. Finally, like DISCO, [77] compares to defenses in RobustBench. For Cifar10 and a WRN28-10 classifier, [77] achieves 70.64/78.58 RA under $\epsilon_\infty = 8/255$ and $\epsilon_2 = 0.5$ respectively, while DISCO achieves 85.56/88.47 (Table 1 & Table 2). On ImageNet, [77] achieves 40.93/44.39 RA with ResNet50/WRN50, while DISCO achieves 68.2/69.5 (Appendix Table D). In summary, DISCO outperforms all these approaches in the various settings they considered, frequently achieving large gains in RA.

**Dataset Size:** Table 4 shows the SA and RA performance of DISCO when training pairs are sampled from a random subset of the ImageNet classes (100 and 500). Compared to the ImageNet SOTA [95] RA of 38.14% (See Appendix), DISCO outperforms the prior art by 21.7% (59.84 vs 38.14) when trained on about 0.5% of the ImageNet training data.

**Defense Transferability** The transferability of the DISCO defense is investigated across attacks, classifiers and datasets.

*Transfer across Attacks.* RobustBench evaluates the model on Autoattack, which includes the PGD attack used to train DISCO. Table 6 summarizes the transfer performance of DISCO, trained with PGD attacks, when subject to ten different $L_\infty$ attacks at inference. This is compared to the transfer peformance of the two top Cifar100 baselines on RobustBench. DISCO outperforms these baselines with an average RA gain greater than 24.49%. When compared to the baseline that uses the same classifier (WRN28-10), this gain increases to 29.5%. Fig. 7 visualizes the gains of DISCO (red bar) on Cifar10 and ImageNet. Among 3 datasets and 10 attacks, DISCO outperforms the baselines on 24 results. The average gains are largest on Cifar100 and ImageNet, where the RA of the prior approaches is lower. Note that the defense is more challenging on ImageNet, due to the higher dimensionality of its images [99]. The full table can be found in the supplemental.

We next evaluate the transfer ability of DISCO trained with the $L_\infty$ PGD attacks to four $L_2$ norm inference attacks: FGSM [33], BIM [60], CW [12] and DeepFool [75]. Table 5 compares the defense transferability of DISCO to both adversarially trained (top block) and transformation baselines (lower block). DISCO generalizes well to $L_2$ attacks. It can defend more attacks than adversarially trained baselines (top block) and is more robust than the prior SOTA transformation based defenses.

Beyond different test attacks $A_{tst}$ on a PGD-trained DISCO, we also evaluated the effect of changing the training attack $A_{trn}$ used to generate the adversarial-clean pairs on which DISCO is trained. In

| Method | Gowal [34] | Rebuffi [88] | DISCO |
|---|---|---|---|
| Classifier | WRN70-16 | WRN28-10 | WRN28-10 |
| FGSM [33] | 44.53 | 38.57 | **50.4** |
| PGD [69] | 40.46 | 36.09 | **74.51** |
| BIM [60] | 40.38 | 36.03 | **72.25** |
| RFGSM [112] | 40.42 | 35.99 | **72.1** |
| EotPgd [67] | 41.07 | 36.45 | **74.8** |
| TPgd [134] | 57.52 | 52.01 | **74.06** |
| FFgsm [119] | 47.61 | 41.47 | **64.29** |
| MiFgsm [24] | 42.37 | 37.31 | **44.14** |
| APgd [20] | 39.99 | 35.64 | **77.33** |
| Jitter [97] | 38.38 | 33.04 | **73.75** |
| Avg. | 43.27 | 38.26 | **67.76** |

Table 6: Defense transfer across ten $L_\infty$ attacks ($\epsilon_\infty = 8/255$) on Cifar100.

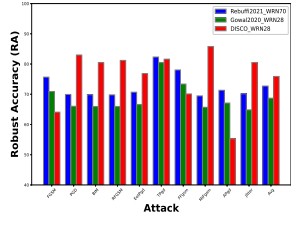
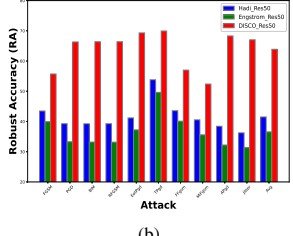

(a)        (b)

Figure 7: Defense transfer across $L_\infty$ attacks on (a) Cifar10 and (b) ImageNet. (Blue, Green) Baselines, (Red) DISCO.

Table 7: Defense transfer of DISCO across training attacks, classifiers, and datasets. In all cases the inference setting is: Cifar10 dataset with Autoattack. For comparison, the RobustBench SOTA [88] for no transfer is also shown.

| | Transfer | | | Training | | | Testing | | | |
|---|---|---|---|---|---|---|---|---|---|---|
| | Classifier | Attack | Dataset | Attack | Classifier | Dataset | Classifier | SA | RA | Avg. |
| [88] | | | | Autoattack | WRN70-16 | Cifar10 | WRN70-16 | 92.23 | 66.58 | 79.41 |
| DISCO | | | | PGD | Res18 | Cifar10 | Res18 | 89.57 | 76.03 | 82.8 |
| | ✓ | | | PGD | Res18 | Cifar10 | VGG16 | 89.12 | 86.27 | 87.7 |
| | ✓ | | | PGD | Res18 | Cifar10 | WRN28 | 89.26 | 85.56 | 87.41 |
| | ✓ | ✓ | | BIM | Res18 | Cifar10 | WRN28 | 91.96 | 84.92 | 88.44 |
| | ✓ | ✓ | | FGSM | Res18 | Cifar10 | WRN28 | 84.07 | 77.13 | 80.6 |
| | ✓ | ✓ | ✓ | FGSM | Res18 | Cifar100 | WRN28 | 84.23 | 86.16 | 85.2 |
| | ✓ | ✓ | ✓ | FGSM | Res18 | ImageNet | WRN28 | 88.91 | 74.3 | 81.61 |

rows 3-5 of Table 7, PGD [69], BIM [60] and FGSM [33] are used to generate training pairs, while Autoattack is used as testing attack. BIM and PGD have comparable results, which are stronger than those of FGSM. Nevertheless all methods outperform the SOTA RobustBench defense [88] for Autoattack on Cifar10, shown in the first row. These results suggests that DISCO is robust to many combinations of training and inference attacks.

***Transfer across Classifiers.*** The first section of Table 7 shows the results when the testing classifier is different from the training classifier. While the ResNet18 is always used to curate the training pairs of DISCO, the testing classifier varies between ResNet18, WideResNet28 and VGG16. The small impact of the classifier used for inference on the overall RA shows that DISCO is classifier agnostic and can be applied to multiple testing classifiers once it is trained.

***Transfer across Datasets.*** The evidence that adversarial attacks push images away from the natural image manifold [135, 64, 107, 27, 47, 66] and that attacks can be transferred across classifiers [22, 117, 121, 45, 116], suggest that it may be possible to transfer defenses across datasets. This, however, has not been studied in detail in the literature, partly because adversarially trained baselines entangle the defense and the classifier training. This is unlike DISCO and other transformation based baselines, which can be transferred across datasets. The bottom section of Table 7 shows the test performance on Cifar10 of DISCO trained on Cifar100 and ImageNet. Since Cifar100 images are more similar to those of Cifar10 than ImageNet ones, the Cifar100 trained DISCO transfers better than that trained on ImageNet. However, even the RA of the latter is 7.72% higher than the best RA reported on RobustBench [88]. Note that the DISCO trained on Cifar100 and ImageNet never see images from Cifar10 and the transfer is feasible because no limitation is imposed on the output size of DISCO.

## 4.2 Adaptive Adversary

The adaptive adversary assumes both the classifier and defense strategy are exposed to the attacker. As noted by [5, 107, 111], this setting is more challenging, especially for transformation based defenses. We adopt the BPDA [5] attack, which is known as an effective attack for transformation based defenses, such as DISCO. Fig. 8 compares the RA of DISCO trained with PGD attack to the results published for other methods in [107]. For fair comparison, DISCO is combined with a VGG16 classifier. The figure confirms that both prior transformation defenses and a single stage DISCO ($K = 1$) are vulnerable to an adaptive adversary. However, without training against BPDA, DISCO is 46.76% better than prior methods. More importantly, this gain increases substantially with $K$, saturating at RA of 57.77% for $K = 5$ stages, which significantly outperforms the SOTA by 57.35%.

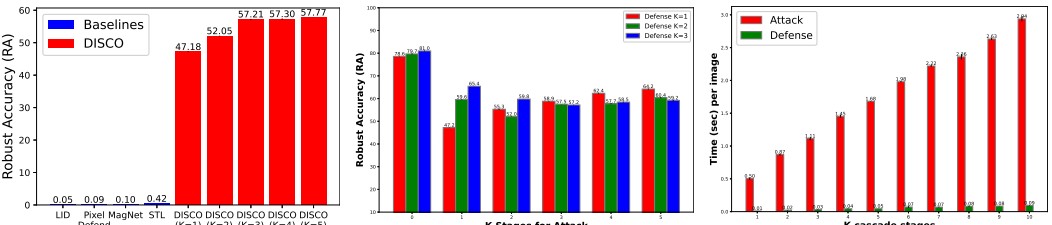

Figure 8: BPDA attack on Cifar10 under the adaptive setting.

Figure 9: BPDA attack with cascade DISCO on Cifar10.

Figure 10: Attack and defense times vs DISCO cascade length.

### 4.3 Cascade DISCO

So far, we have considered the setting where the structure of the cascade DISCO is known to the attacker. DISCO supports a more sophisticated and practical setting, where the number $K$ of DISCO stages used by the defense is randomized on a per-image basis. In this case, even if the use of DISCO is exposed to the attacker, there is still uncertainty about how many stages to use in the attack. We investigated the consequences of this uncertainty by measuring the defense performance when different values of $K$ are use for attack and defense, denoted as $K_{adv}$ and $K_{def}$, respectively. The oblivious setting has $K_{adv} = 0$ and $K_{def} \geq 1$, while $K_{adv} = K_{def}$ in the adaptive setting. We now consider the case where $K_{adv} \neq K_{def}$. Fig. 9 investigates the effectiveness of cascade DISCO trained with PGD attack when faced with the BPDA [5] attack in this setting, where RA($K_{adv}$,$K_{def}$) is the RA when $K_{adv}$ and $K_{def}$ are used, and $K_{adv} \in \{i\}_{i=0}^{5}$, $K_{def} \in \{i\}_{i=1}^{3}$. Under the setting of $K_{adv} \neq K_{def}$, the RA is higher than that of the adaptive setting. Take $K_{adv} = 2$ for example. Both RA(2,1)=55.3 and RA(2,3)=59.8 outperform RA(2,2)=52. In addition, Fig. 10 compares the time to generate a single adversarial example on Cifar10 and defend against it using DISCO. Clearly, the computational resources needed to generate an attack are significantly higher than those of the defense and the ratio of attack-to-defense cost raises with $K$. Both this and the good defense performance for mismatched $K$s give the defender a strong advantage. It appears that the defense is more vulnerable when the attacker knows $K$ (adaptive setting) and even there, as seen in the previous section, the defense can obtain the upper hand by casacading several DISCO stages.

## 5 Discussion, Societal Impact and Limitations

In this work, we have proposed the use of local implicit functions for adversarial defense. Given an input adversarial image and a query location, the DISCO model is proposed to project the RGB value of each image pixel into the image manifold, conditional on deep features centered in the pixel neighborhood. By training this projection with adversarial and clean images, DISCO learns to remove the adversarial perturbation. Experiments demonstrate DISCO's computational efficiency, its outstanding defense performance and transfer ability across attacks, datasets and classifiers. The cascaded version of DISCO further strengthens the defense with minor additional cost.

**Limitations:** While DISCO shows superior performance on the attacks studied in this work (mainly norm-bounded attacks), it remains to be tested whether it is robust to other type of attacks [106, 11, 42, 40, 61], such as one pixel attack [106], patch attacks [11, 42] or functional adversarial attack [61]. In addition, more evaluation configurations across attacks, datasets and classifiers will be investigated in the future.

**Societal Impact:** We hope the idea of using local implicit functions can inspire better defenses and prevent the nefarious effects of deep learning attacks. Obviously, better defenses can also be leveraged by bad actors to improve resistance to the efforts of law enforcement, for example.

## 6 Acknowledgments

This work was partially funded by NSF awards IIS1924937 and IIS-2041009, a gift from Amazon, a gift from Qualcomm, and NVIDIA GPU donations. We also acknowledge and thank the use of the Nautilus platform for some of the experiments discussed above.

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
