# DISCO: Adversarial Defense with Local Implicit Functions

**Chih-Hui Ho    Nuno Vasconcelos**
Department of Electrical and Computer Engineering
University of California, San Diego
{chh279, nvasconcelos}@ucsd.edu

## A    Compare to SOTA in RobustBench

In this section, we list the quantitative result of the baselines in RobustBench [14] . Table A, C and D correspond to Fig.6(a), (b) and (c) of the main paper, respectively. Table B shows the baselines under Autoattack with $\epsilon_2 = 0.5$. The index displayed in each table corresponds to the index shown in Fig.6 in the main paper. The baselines of each table are grouped into No defense (first block), Adversarially trained defense in RobustBench (second block), Transformation based defense (third block) and DISCO (last block). The results of adversarially trained baselines are copied from RobustBench, while the results of transformation-based defenses are obtained with our implementation. For STL [60], models with different sparse constraints $\lambda$ are used from the publicly available STL github[1]. DISCO is also combined with various classifiers for evaluation. More discussion can be found in Sec. 4.1 of the paper.

| ID | Method | Standard Acc. | Robust Acc. | Avg. Acc. | Model | ID | Method | Standard Acc. | Robust Acc. | Avg. Acc. | Model |
|---|---|---|---|---|---|---|---|---|---|---|---|
| 0 | No Defense | 94.78 | 0 | 47.39 | WRN28-10 | | | | | | |
| 1 | Rebuffi et al. [48] | 92.23 | 66.58 | 79.41 | WRN70-16 | 2 | Gowal et al. [25] | 88.74 | 66.11 | 77.43 | WRN70-16 |
| 3 | Gowal et al. [24] | 91.1 | 65.88 | 78.49 | WRN70-16 | 4 | Rebuffi et al. [48] | 88.5 | 64.64 | 76.57 | WRN106-16 |
| 5 | Rebuffi et al. [48] | 88.54 | 64.25 | 76.4 | WRN70-16 | 6 | Kang et al. [31] | 93.73 | 71.28 | 82.51 | WRN70-16 |
| 7 | Gowal et al. [25] | 87.5 | 63.44 | 75.47 | WRN28-10 | 8 | Pang et al. [41] | 89.01 | 63.35 | 76.18 | WRN70-16 |
| 9 | Rade et al. [47] | 91.47 | 62.83 | 77.15 | WRN34-10 | 10 | Sehwag et al. [53] | 87.3 | 62.79 | 75.05 | ResNest152 |
| 11 | Gowal et al. [24] | 89.48 | 62.8 | 76.14 | WRN28-10 | 12 | Huang et al. [27] | 91.23 | 62.54 | 76.89 | WRN34-R |
| 13 | Huang et al. [27] | 90.56 | 61.56 | 76.06 | WRN34-R | 14 | Dai et al. [18] | 87.02 | 61.55 | 74.29 | WRN28-10 |
| 15 | Pang et al. [41] | 88.61 | 61.04 | 74.83 | WRN28-10 | 16 | Rade et al. [47] | 88.16 | 60.97 | 74.57 | WRN28-10 |
| 17 | Rebuffi et al. [48] | 87.33 | 60.75 | 74.04 | WRN28-10 | 18 | Wu et al. [66] | 87.67 | 60.65 | 74.16 | WRN34-15 |
| 19 | Sridhar et al. [59] | 86.53 | 60.41 | 73.47 | WRN34-15 | 20 | Sehwag et al. [54] | 86.68 | 60.27 | 73.48 | WRN34-10 |
| 21 | Wu et al. [67] | 88.25 | 60.04 | 74.15 | WRN28-10 | 22 | Sehwag et al. [54] | 89.46 | 59.66 | 74.56 | WRN28-10 |
| 23 | Zhang et al. [77] | 89.36 | 59.64 | 74.5 | WRN28-10 | 24 | Yair et al. [8] | 89.69 | 59.53 | 74.61 | WRN28-10 |
| 25 | Gowal et al. [25] | 87.35 | 58.63 | 72.99 | PreActRes18 | 26 | Addepalli et al. [1] | 85.32 | 58.04 | 71.68 | WRN34-10 |
| 27 | Chen et al. [10] | 86.03 | 57.71 | 71.87 | WRN34-20 | 28 | Rade et al. [47] | 89.02 | 57.67 | 73.35 | PreActRes18 |
| 29 | Gowal et al. [24] | 85.29 | 57.2 | 71.25 | WRN70-16 | 30 | Sehwag et al. [55] | 88.98 | 57.14 | 73.06 | WRN28-10 |
| 31 | Rade et al. [47] | 86.86 | 57.09 | 71.98 | PreActRes18 | 32 | Chen et al. [10] | 85.21 | 56.94 | 71.08 | WRN34-10 |
| 33 | Gowal et al. [24] | 85.64 | 56.86 | 71.25 | WRN34-20 | 34 | Rebuffi et al. [48] | 83.53 | 56.66 | 70.1 | PreActRes18 |
| 35 | Wang et al. [63] | 87.5 | 56.29 | 71.9 | WRN28-10 | 36 | Wu et al. [67] | 85.36 | 56.17 | 70.77 | WRN34-10 |
| 37 | Alayrac et al. [3] | 86.46 | 56.03 | 71.25 | WRN28-10 | 38 | Sehwag et al. [54] | 84.59 | 55.54 | 70.07 | Res18 |
| 39 | Dan et al. [26] | 87.11 | 54.92 | 71.02 | WRN28-10 | 40 | Pang et al. [43] | 86.43 | 54.39 | 70.41 | WRN34-20 |
| 41 | Pang et al. [44] | 85.14 | 53.74 | 69.44 | WRN34-20 | 42 | Cui et al. [17] | 88.7 | 53.57 | 71.14 | WRN34-20 |
| 43 | Zhang et al. [76] | 84.52 | 53.51 | 69.02 | WRN34-10 | 44 | Rice et al. [49] | 85.34 | 53.42 | 69.38 | WRN34-20 |
| 45 | Huang et al. [28] | 83.48 | 53.34 | 68.41 | WRN34-10 | 46 | Zhang et al. [74] | 84.92 | 53.08 | 69 | WRN34-10 |
| 47 | Cui et al. [16] | 88.22 | 52.86 | 70.54 | WRN34-10 | 48 | Qin et al. [46] | 86.28 | 52.84 | 69.56 | WRN40-8 |
| 49 | Chen et al. [12] | 86.04 | 51.56 | 68.8 | Res50 | 50 | Chen et al. [11] | 85.32 | 51.12 | 68.22 | WRN34-10 |
| 51 | Addepalli et al. [2] | 80.24 | 51.06 | 65.65 | Res18 | 52 | Chawin et al. [58] | 86.84 | 50.72 | 68.78 | WRN34-10 |
| 53 | Engstrom et al. [22] | 87.03 | 49.25 | 68.14 | Res50 | 54 | Sinha et al. [57] | 87.8 | 49.12 | 68.46 | WRN34-10 |
| 55 | Mao et al. [38] | 86.21 | 47.41 | 66.81 | WRN34-10 | 56 | Zhang et al. [71] | 87.2 | 44.83 | 66.02 | WRN34-10 |
| 57 | Madry et al. [36] | 87.14 | 44.04 | 65.59 | WRN34-10 | 58 | Maksym et al. [4] | 79.84 | 43.93 | 61.89 | PreActRes18 |
| 59 | Pang et al. [42] | 80.89 | 43.48 | 62.19 | Res32 | 60 | Wong et al. [64] | 83.34 | 43.21 | 63.28 | PreActRes18 |
| 61 | Shafahi et al. [56] | 86.11 | 41.47 | 63.79 | WRN34-10 | 62 | Ding et al. [19] | 84.36 | 41.44 | 62.9 | WRN28-4 |
| 63 | Souvik et al. [33] | 87.32 | 40.41 | 63.87 | Res18 | 64 | Matan et al. [6] | 81.3 | 40.22 | 60.76 | Res18 |
| 65 | Moosavi-Dezfooli et al. [39] | 83.11 | 38.5 | 60.81 | Res18 | 66 | Zhang et al. [72] | 89.98 | 36.64 | 63.31 | WRN28-10 |
| 67 | Zhang et al. [73] | 90.25 | 36.45 | 63.35 | WRN28-10 | 68 | Jang et al. [29] | 78.91 | 34.95 | 56.93 | Res20 |
| 69 | Kim et al. [32] | 91.51 | 34.22 | 62.87 | WRN34-10 | 70 | Zhang et al. [75] | 44.73 | 32.64 | 38.69 | 5 layer CNN |
| 71 | Wang et al. [62] | 92.8 | 29.35 | 61.08 | WRN28-10 | 72 | Xiao et al. [68] | 79.28 | 18.5 | 48.89 | DenseNet121 |
| 73 | Jin et al. [30] | 90.84 | 1.35 | 46.1 | Res18 | 74 | Aamir et al. [40] | 89.16 | 0.28 | 44.72 | Res110 |
| 75 | Chan et al. [9] | 93.79 | 0.26 | 47.03 | WRN34-10 | | | | | | |
| 76 | Bit Reduction [70] | 92.66 | 1.04 | 46.85 | WRN28-10 | 77 | Jpeg [21] | 83.9 | 50.73 | 67.32 | WRN28-10 |
| 78 | Input Rand. [69] | 94.3 | 8.59 | 51.45 | WRN28-10 | 79 | LIIF [13] | 94.85 | 0.22 | 47.54 | WRN28-10 |
| 80 | AutoEncoder | 76.54 | 67.41 | 71.98 | WRN28-10 | 81 | STL [60] (k=64 s=8 λ=0.1) | 90.65 | 57.28 | 73.97 | WRN28-10 |
| 82 | STL [60] (k=64 s=8 λ=0.15) | 86.77 | 66.94 | 76.86 | WRN28-10 | 83 | STL [60] (k=64 s=8 λ=0.2) | 82.22 | 67.92 | 75.07 | WRN28-10 |
| 84 | Median Filter | 79.67 | 42.49 | 61.08 | WRN28-10 | | | | | | |
| 85 | DISCO | 89.26 | 85.56 ± 0.02 | 87.41 | WRN28-10 | | | | | | |

Table A: Cifar10 baselines and DISCO under Autoattack ($\epsilon_\infty = 8/255$). This table corresponds to Fig. 6(a) in the main paper.

| ID | Method | Standard Acc. | Robust Acc. | Avg. Acc. | Model | ID | Method | Standard Acc. | Robust Acc. | Avg. Acc. | Model |
|---|---|---|---|---|---|---|---|---|---|---|---|
| 0 | No Defense | 94.78 | 0 | 47.39 | WRN28-10 | | | | | | |
| 1 | Rebuffi et al. [48] | 95.74 | 82.32 | 89.03 | WRN70-16 | 2 | Gowal et al. [24] | 94.74 | 80.53 | 87.64 | WRN70-16 |
| 3 | Rebuffi et al. [48] | 92.41 | 80.42 | 86.42 | WRN70-16 | 4 | Rebuffi et al. [48] | 91.79 | 78.8 | 85.30 | WRN28-10 |
| 5 | Augustin et al. [7] | 93.96 | 78.79 | 86.38 | WRN34-10 | 6 | Sehwag et al. [53] | 90.93 | 77.24 | 84.09 | WRN34-10 |
| 7 | Augustin et al. [7] | 92.23 | 76.25 | 84.24 | WRN34-10 | 8 | Rade et al. [47] | 90.57 | 76.15 | 83.36 | PreActRes18 |
| 9 | Rebuffi et al. [48] | 90.33 | 75.86 | 83.10 | PreActRes18 | 10 | Gowal et al. [24] | 90.9 | 74.5 | 82.70 | WRN70-16 |
| 11 | Sehwag et al. [53] | 89.76 | 74.41 | 82.09 | Res18 | 12 | Wu et al. [67] | 88.51 | 73.66 | 81.09 | WRN34-10 |
| 13 | Augustin et al. [7] | 91.08 | 72.91 | 82.00 | Res50 | 14 | Engstrom et al. [22] | 90.83 | 69.24 | 80.04 | Res50 |
| 15 | Rice et al. [49] | 88.67 | 67.68 | 78.18 | PreActRes18 | 16 | Rony et al. [50] | 89.05 | 66.44 | 77.75 | WRN28-10 |
| 17 | Ding et al. [19] | 88.02 | 66.09 | 77.06 | WRN28-4 | | | | | | |
| 18 | Bit Reduction [70] | 92.66 | 3.8 | 48.23 | WRN28-10 | 19 | Jpeg [21] | 83.9 | 69.85 | 76.88 | WRN28-10 |
| 20 | Input Rand. [69] | 94.3 | 25.71 | 60.01 | WRN28-10 | 21 | LIIF [13] | 94.85 | 0.22 | 47.54 | WRN28-10 |
| 22 | AutoEncoder | 76.54 | 71.71 | 74.13 | WRN28-10 | 23 | STL [60] (k=64 s=8 λ=0.1) | 90.65 | 75.55 | 83.1 | WRN28-10 |
| 24 | STL [60] (k=64 s=8 λ=0.15) | 86.77 | 76.45 | 81.61 | WRN28-10 | 25 | STL [60] (k=64 s=8 λ=0.2) | 82.22 | 74.33 | 78.28 | WRN28-10 |
| 26 | Median Filter | 79.67 | 63.94 | 71.81 | WRN28-10 | | | | | | |
| 27 | DISCO | 89.26 | 88.47 ± 0.16 | 88.87 | WRN28-10 | | | | | | |

Table B: Cifar10 baselines and DISCO under Autoattack ($\epsilon_2 = 0.5$).

| ID | Method | Standard Acc. | Robust Acc. | Avg. Acc. | Model | ID | Method | Standard Acc. | Robust Acc. | Avg. Acc. | Model |
|---|---|---|---|---|---|---|---|---|---|---|---|
| 0 | No Defense | 80.37 | 0 | 41.78 | WRN28-10 | | | | | | |
| 1 | Gowal et al. [24] | 69.15 | 36.88 | 53.02 | WRN70-16 | 2 | Rebuffi et al. [48] | 63.56 | 34.64 | 49.1 | WRN70-16 |
| 3 | Pang et al. [41] | 65.56 | 33.05 | 49.31 | WRN70-16 | 4 | Rebuffi et al. [48] | 62.41 | 32.06 | 47.24 | WRN28-10 |
| 5 | Sehwag et al. [53] | 65.93 | 31.15 | 48.54 | WRN34-10 | 6 | Pang et al. [41] | 63.66 | 31.08 | 47.37 | WRN28-10 |
| 7 | Chen et al. [10] | 64.07 | 30.59 | 47.33 | WRN34-10 | 8 | Addepalli et al. [2] | 65.73 | 30.35 | 48.04 | WRN34-10 |
| 9 | Cui et al. [17] | 62.55 | 30.2 | 46.38 | WRN34-20 | 10 | Gowal et al. [24] | 60.86 | 30.03 | 45.45 | WRN70-16 |
| 11 | Cui et al. [17] | 60.64 | 29.33 | 44.99 | WRN34-10 | 12 | Rade et al. [47] | 61.5 | 28.88 | 45.19 | PreActRes18 |
| 13 | Wu et al. [67] | 60.38 | 28.86 | 44.62 | WRN34-10 | 14 | Rebuffi et al. [48] | 56.87 | 28.5 | 42.69 | PreActRes18 |
| 15 | Dan et al. [26] | 59.23 | 28.42 | 43.83 | WRN28-10 | 16 | Cui et al. [17] | 70.25 | 27.16 | 48.71 | WRN34-10 |
| 17 | Addepalli et al. [2] | 62.02 | 27.14 | 44.58 | PreActRes18 | 18 | Chen et al. [11] | 62.15 | 26.94 | 44.55 | WRN34-10 |
| 19 | Chawin et al. [58] | 62.82 | 24.57 | 43.7 | WRN34-10 | 20 | Rice et al. [49] | 53.83 | 18.95 | 36.39 | PreActRes18 |
| 21 | Bit Reduction [70] | 76.86 | 3.78 | 40.32 | WRN28-10 | 22 | Jpeg [21] | 61.89 | 39.59 | 50.74 | WRN28-10 |
| 23 | Input Rand. [69] | 73.57 | 3.31 | 38.44 | WRN28-10 | 24 | LIIF [13] | 80.3 | 3.36 | 41.83 | WRN28-10 |
| 25 | AutoEncoder | 58.79 | 48.36 | 53.575 | WRN28-10 | 26 | STL [60] (k=64 s=8 λ=0.1) | 74.28 | 30.05 | 52.17 | WRN28-10 |
| 27 | STL [60] (k=64 s=8 λ=0.15) | 70.3 | 41.82 | 56.06 | WRN28-10 | 28 | STL [60] (k=64 s=8 λ=0.2) | 67.41 | 46.07 | 56.74 | WRN28-10 |
| 29 | Median Filter | 65.78 | 34.52 | 50.15 | WRN28-10 | | | | | | |
| 30 | DISCO | 72.07 | 67.93±0.17 | 70 | WRN28-10 | 31 | DISCO | 71.62 | 69.01 ±0.19 | 70.32 | WRN34-10 |

Table C: Cifar100 baselines and DISCO under Autoattack ($\epsilon_\infty = 8/255$). This table corresponds to Fig. 6(b) in the main paper.

| ID | Method | Standard Acc. | Robust Acc. | Avg. Acc. | Model | ID | Method | Standard Acc. | Robust Acc. | Avg. Acc. | Model |
|---|---|---|---|---|---|---|---|---|---|---|---|
| 0 | No Defense | 76.52 | 0 | 38.26 | Res50 | | | | | | |
| 1 | Hadi et al. [51] | 68.46 | 38.14 | 53.3 | WRN50-2 | 2 | Hadi et al. [51] | 64.02 | 34.96 | 49.49 | Res50 |
| 3 | Engstrom et al. [22] | 62.56 | 29.22 | 45.89 | Res50 | 4 | Wong et al. [64] | 55.62 | 26.24 | 40.93 | Res50 |
| 5 | Hadi et al. [51] | 52.92 | 25.32 | 39.12 | Res18 | | | | | | |
| 6 | Bit Reduction [70] | 67.64 | 4.04 | 35.84 | Res18 | 7 | Bit Reduction [70] | 73.82 | 1.86 | 37.84 | Res50 |
| 8 | Bit Reduction [70] | 75.06 | 4.96 | 40.01 | WRN50-2 | 9 | Jpeg [21] | 67.18 | 13.08 | 40.13 | Res18 |
| 10 | Jpeg [21] | 73.64 | 33.42 | 53.53 | Res50 | 11 | Jpeg [21] | 75.42 | 24.9 | 50.16 | WRN50-2 |
| 12 | Input Rand. [69] | 64 | 17.78 | 40.89 | Res18 | 13 | Input Rand.. [69] | 74.02 | 18.84 | 46.43 | Res50 |
| 14 | Input Rand. [69] | 71.7 | 23.58 | 47.64 | WRN50-2 | 15 | STL [60] (k=64 s=8 λ=0.1) | 67.5 | 18.5 | 43 | Res18 |
| 16 | STL [60] (k=64 s=8 λ=0.2) | 65.64 | 32.9 | 49.27 | Res18 | 17 | STL [60] (k=64 s=8 λ=0.1) | 72.56 | 32.7 | 52.63 | Res50 |
| 18 | STL [60] (k=64 s=8 λ=0.2) | 68.3 | 50.16 | 59.23 | Res50 | 19 | Median Filter | 66.1 | 10.34 | 38.22 | Res18 |
| 20 | Median Filter | 71.68 | 17.36 | 44.52 | Res50 | | | | | | |
| 21 | DISCO | 67.98 | 60.88±0.17 | 64.43 | Res18 | 22 | DISCO | 72.64 | 68.2±0.29 | 70.42 | Res50 |
| 23 | DISCO | 75.1 | 69.5±0.23 | 72.3 | WRN50-2 | | | | | | |

Table D: ImageNet baselines and DISCO under Autoattack ($\epsilon_\infty = 4/255$). This table corresponds to Fig. 6(c) in the main paper.

---

[1] https://github.com/GitBoSun/AdvDefense_CSC

# B Defense Transfer

In this section, we discuss the qualitative results of DISCO transferability across attacks. Table E, F and G represents the results for Cifar10, Cifar100 and ImageNet, respectively. The corresponding plots are illustrated in Fig. A, B and C. More discussion can be found in Sec. 4.1 of the paper.

Table E: Defense Transfer across $L_\infty$ attacks ($\epsilon_\infty = 8/255$) on Cifar10.

| Method | Rebuffi et al. [48] | Gowal et al. [24] | DISCO |
|---|---|---|---|
| Classifier | WRN70-16 | WRN28-10 | WRN28-10 |
| FGSM [23] | **75.66** | 70.91 | 64.08 |
| PGD [37] | 69.93 | 66.02 | **82.99** |
| BIM [34] | 69.84 | 65.95 | **80.46** |
| RFGSM [61] | 69.8 | 65.95 | **81.2** |
| EotPgd [35] | 70.68 | 66.58 | **76.84** |
| TPgd [74] | **82.32** | 80.48 | 81.61 |
| FFgsm [65] | **78.04** | 73.37 | 70.1 |
| MiFgsm [20] | **73.22** | 68.82 | 45.49 |
| APgd [15] | 69.46 | 65.69 | **85.79** |
| Jitter [52] | 70.15 | 64.84 | **80.49** |
| Avg. | 72.72 | 68.69 | **75.88** |

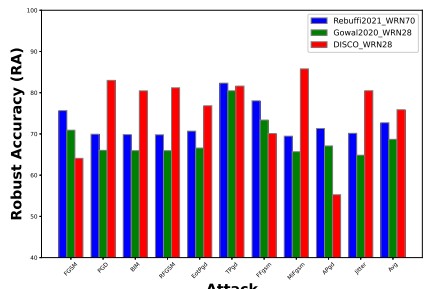

Figure A: Defense Transfer across $L_\infty$ attacks on Cifar10.

Table F: Defense Transfer across $L_\infty$ attacks ($\epsilon_\infty = 8/255$) on Cifar100.

| Method | Gowal et al. [24] | Rebuffi et al. [48] | DISCO |
|---|---|---|---|
| Classifier | WRN70-16 | WRN28-10 | WRN28-10 |
| FGSM [23] | 44.53 | 38.57 | **50.4** |
| PGD [37] | 40.46 | 36.09 | **74.51** |
| BIM [34] | 40.38 | 36.03 | **72.25** |
| RFGSM [61] | 40.42 | 35.99 | **72.1** |
| EotPgd [35] | 41.07 | 36.45 | **74.8** |
| TPgd [74] | 57.52 | 52.01 | **74.06** |
| FFgsm [65] | 47.61 | 41.47 | **64.29** |
| MiFgsm [20] | 42.37 | 37.31 | **44.14** |
| APgd [15] | 39.99 | 35.64 | **77.33** |
| Jitter [52] | 38.38 | 33.04 | **73.75** |
| Avg. | 43.27 | 38.26 | **67.76** |

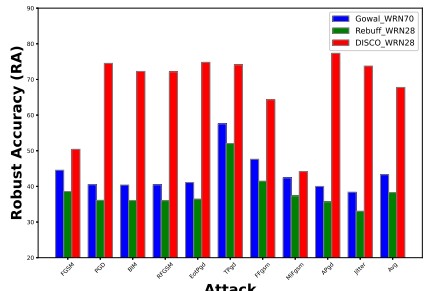

Figure B: Defense Transfer across $L_\infty$ attacks on Cifar100.

Table G: Defense Transfer across $L_\infty$ attacks ($\epsilon_\infty = 4/255$) on ImageNet.

| Method | Hadi et al. [51] | Engstrom et al. [22] | DISCO |
|---|---|---|---|
| Classifier | Res50 | Res50 | Res50 |
| Clean | 64.1 | 62.54 | **72.64** |
| FGSM [23] | 43.48 | 39.96 | **55.72** |
| PGD [37] | 39.28 | 33.32 | **66.32** |
| BIM [34] | 39.26 | 33.2 | **66.4** |
| RFGSM [61] | 39.28 | 33.16 | **66.4** |
| EotPgd [35] | 41.2 | 37.24 | **69.32** |
| TPgd [74] | 53.82 | 49.64 | **69.94** |
| FFgsm [65] | 43.58 | 40.1 | **57** |
| MiFgsm [20] | 40.56 | 35.6 | **52.38** |
| APgd [15] | 38.42 | 32.22 | **68.3** |
| Jitter [52] | 36.26 | 31.36 | **67.04** |
| Avg. | 41.51 | 36.58 | **63.88** |

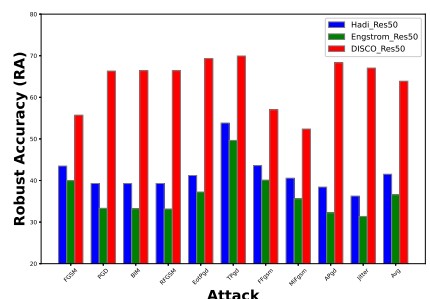

Figure C: Defense Transfer across $L_\infty$ attacks on ImageNet.

# C Improving Cifar10 and Cifar100 SOTA on RobustBench

Sec. 4.1 in the main paper shows that DISCO can improve the prior SOTA defenses on the ImageNet dataset. In Table H, we further investigate the gain of applying DISCO on SOTA Cifar10 and Cifar100 defenses. The first and second block of Table H show the gains of applying DISCO on [48], which is the prior SOTA defense against $L_2$ and $L_\infty$ Autoattack on Cifar10. DISCO also improves the prior

Table H: Improving SOTA defenses on RobustBench [14] for Cifar10 ($L_2$ and $L_\infty$) and Cifar100 ($L_\infty$) dataset.

| Method | Dataset | Norm | SA | RA | Avg. |
|---|---|---|---|---|---|
| Rebuffi et al. [48] | Cifar10 | $L_\infty$ | **92.23** | 66.58 | 79.41 |
| w/ DISCO | Cifar10 | $L_\infty$ | 91.95 | **70.71** | **81.33** |
| Rebuffi et al. [48] | Cifar10 | $L_2$ | **95.74** | 82.32 | 89.03 |
| w/ DISCO | Cifar10 | $L_2$ | 95.24 | **84.15** | **89.7** |
| Gowal et al. [24] | Cifar100 | $L_\infty$ | **69.15** | 36.88 | 53.02 |
| w/ DISCO | Cifar100 | $L_\infty$ | 68.56 | **39.77** | **54.17** |

SOTA defense [24] on Cifar100 by 2.89%. These results indicate that, beyond being a robust defense by itself, DISCO can also be applied to existing defenses to improve their robustness.

# D   Kernel Size $s$

Table I: Ablation on various kernel size $s$.

| $s$ | SA | RA | Avg. |
|---|---|---|---|
| 1 | 71.22 | **69.52** | 70.37 |
| 3 | 72.64 | 68.2 | **70.42** |
| 5 | **74.22** | 60.1 | 67.16 |

In this section, we ablate the kernel size used to train DISCO on ImageNet. The kernel size $s$ controls the feature neighborhood forwarded to the local implicit module. Table I shows that $s = 3$ achieves the best performance, which degrades for $s = 5$ by a significant margin (3.26%). This shows that while tasks like classification require large and global receptive fields, the projection of adversarial images into the natural image manifold can be done on small neighborhoods. Given that the complexity of modeling the manifold increases with the neighborhood size, it is not surprising that larger $s$ lead to weaker performance. This is consistent with the well known complexity of synthesizing images with global models, such as GANs. What is somewhat surprising is that even $s = 1$ is sufficient to enable a robust defense. By default, we use $s = 3$ in all our experiments.

# E   Computation Time for STL and DISCO

Table J: Computation time between of STL [60] and DISCO for different image sizes. Note that STL requires a 36.34× larger inference time when image size increases from 32 to 224.

| Dataset | Image Size | STL [60] | DISCO | | | | |
|---|---|---|---|---|---|---|---|
| | | | (K=1) | (K=2) | (K=3) | (K=4) | (K=5) |
| Cifar10 | 32 | 0.65 | 0.011 | 0.021 | 0.031 | 0.037 | 0.048 |
| ImageNet | 224 | 23.71 | 0.027 | 0.081 | 0.134 | 0.191 | 0.251 |
| Time Increase | | ×36.34 | ×2.41 | ×3.86 | ×4.35 | ×5.14 | ×5.19 |

Table J compares the inference time of STL [60], DISCO and cascade DISCO (from $K = 2$ to 5) on Cifar10 and ImageNet. For a single image Cifar10 of size 32x32, STL requires an Cifar10 5.9× (0.65 vs 0.011) larger than that of DISCO ($K$=1). When cascade DISCO is used, inference time increases approximately linearly with $K$.

For a single ImageNet image of size 224, STL requires 23.71 seconds while DISCO (K=1) only requires 0.027. The inference time difference increases to 878.15× (23.71 vs 0.027) on ImageNet, which is significantly larger than that of Cifar10 (5.9×). This shows that DISCO is a better defense in the sense that it can handle widely varying input image sizes with minor variations of computing cost.

## F   Training Details

On Cifar10 and Cifar100, we train the DISCO for 40 epochs. On ImageNet, DISCO is only trained for 3 epochs because ImageNet images are larger and produce more random crops. The learning rate is set to 0.0001 and the Adam optimizer is used in all experiments. All experiments are conducted using Pytorch [45]. All time measurements, for both baselines and DISCO, are made on a single Nvidia Titan Xp GPU with Intel Xeon CPU E5-2630, with batch size 1 and averaged over 100 images.

## G   Adopted Code and Benchmark

In this section, we list the url links that are used for training and evaluating DISCO. To create the adversarial-clean training pairs, we adopt the code from TorchAttack[2] and Ares[3], which support the multiple attack methods. These attack methods are then used to attack pretrained classifiers on Cifar10, Cifar100 and ImageNet. We use the ResNet18 classifiers from Ares[3] for Cifar10, the WideResNet Cifar100 classifiers from this repository [4] and the ResNet18 ImageNet classifiers of Pytorch [45].

To evaluate DISCO, we adopt Autoattack from RobustBench [14][5] and compare to the pretrained defenses on the RobustBench leaderboard. In addition to Autoattack, we use the AdverTorch[6] library to implement the BPDA attack [5] and the TorchAttack[7] library for other attacks, like FGSM [23] and BIM [34].

For the adversarially trained defense baselines, we adopt the pretrained weights from Robust-Bench [14][8], while the codes for transformation based baselines are adopted from Ares[3], Cifar autoencoder [9] and STL[1] [60]. To implement DISCO, we use code from LIIF[10] [13].

## H   DCT Analysis

The effectiveness of perturbation removal can be analyzed in the frequency domain using the discrete cosine transform (DCT). Consider an image $x^i$ and the corresponding clean image $x^i_{cln}$. The average normalized difference (ND) between DCTs over $M$ images is defined as

$$ND(\mathcal{X}) = \log\left(\frac{1}{M}\sum_{i=1}^{M}\left|\frac{DCT(x^i) - DCT(x^i_{Cln})}{DCT(x^i_{Cln})}\right|\right), \tag{1}$$

where $\mathcal{X} = \{x^i\}$ can contain adversarial images $\mathcal{X}_{Adv}$, the outputs of DISCO $\mathcal{X}_{Def}$ or the outputs of cascade DISCO $\mathcal{X}_{CDef}$. Fig. D(a), (b) and (c) shows the ND obtained for $\mathcal{X}_{Adv}$, $\mathcal{X}_{Def}$ and $\mathcal{X}_{CDef}$, for $M = 100$ images. Darker areas indicate higher similarity between clean and input images, at a specific frequency. Take Fig. D(a) for example. The dark area concentrates on the low frequency area (upper left corner), while the bright area concentrates on the high frequency area (lower right corner) showing that adversarial noise is mostly of high frequency. Fig. D(b) shows that, after the adversarial image is forwarded through DISCO, the high frequency area becomes darker. Fig. D(d) further highlights the difference between Fig. D(a) and (b), by illustrating $\mathcal{U}(ND(\mathcal{X}_{Adv}) - ND(\mathcal{X}_{Def}))$, where $\mathcal{U}$ is a unit step function. The white area of Fig. D(d) indicates that most of the high frequency perturbations are removed from the adversarial image by DISCO. Similarly, Fig. D(e) demonstrates that cascade DISCO further removes high frequency perturbations when comparing to Fig D(b) and (c).

---

[2]https://adversarial-attacks-pytorch.readthedocs.io/en/latest/

[3]https://github.com/thu-ml/ares

[4]https://github.com/xternalz/WideResNet-pytorch

[5]https://github.com/RobustBench/robustbench

[6]https://github.com/BorealisAI/advertorch

[7]https://adversarial-attacks-pytorch.readthedocs.io/en/latest/

[8]https://github.com/RobustBench/robustbench

[9]https://github.com/chenjie/PyTorch-CIFAR-10-autoencoder

[10]https://github.com/yinboc/liif

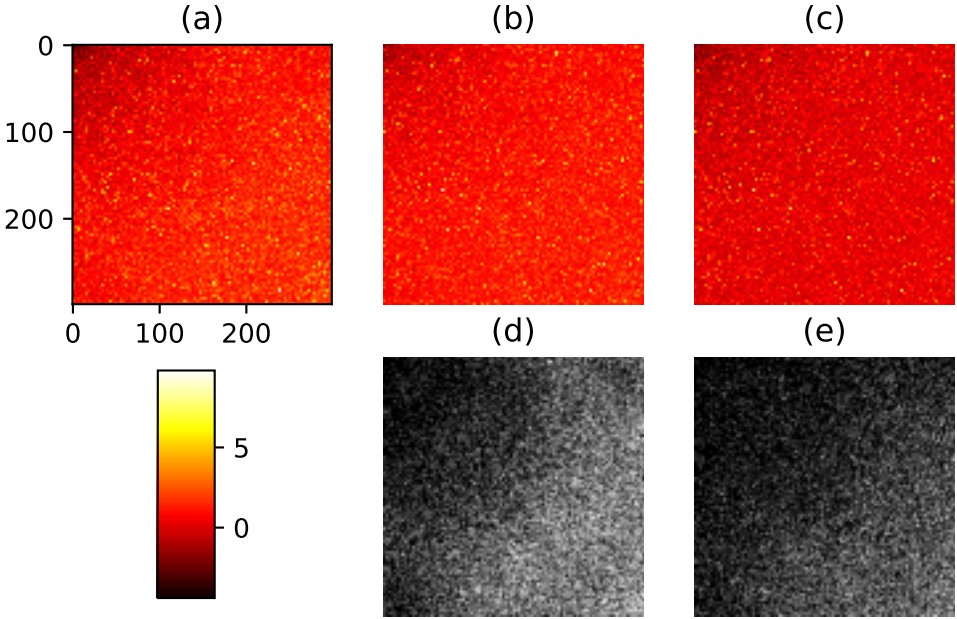

Figure D: (a), (b) and (c) show ND plots for $\mathcal{X}_{Adv}$, $\mathcal{X}_{Def}$ and $\mathcal{X}_{CDef}$, respectively. (d) and (e) show the difference between (a)/(b) and (b)/(c), respectively. See text for more details.

## I    Visualizations

DISCO defense outputs against FGSM [23], BIM [34] and PGD [37] attacks are visualized in Fig. E, F and G, respectively. Take Fig. E for example. The first and second rows show the clean and adversarial images, while rows 3-5 show the output of DISCO and cascade DISCO ($K = 2$ and $K = 3$). Clearly, both DISCO and its cascade version can effectively remove the adversarial perturbation. In addition, Fig. H shows the DISCO output for various images sizes, from 128x128 to 512x512. Note that these images are produced from the same DISCO model without retraining for any output size or attack.

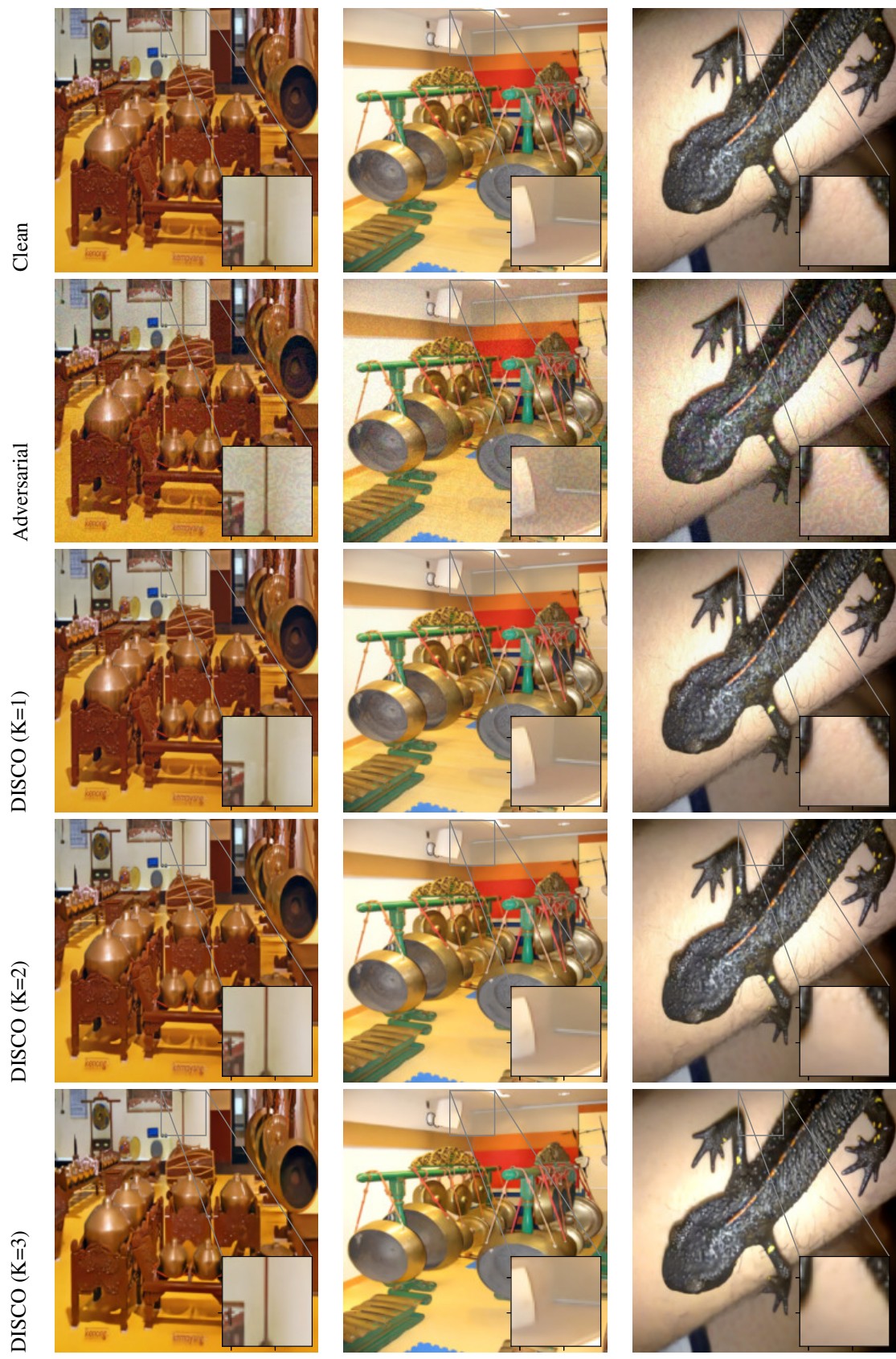

Figure E: Comparison of Clean image, Adversarial image and DISCO output from $K = 1$ to 3 under FGSM attack.

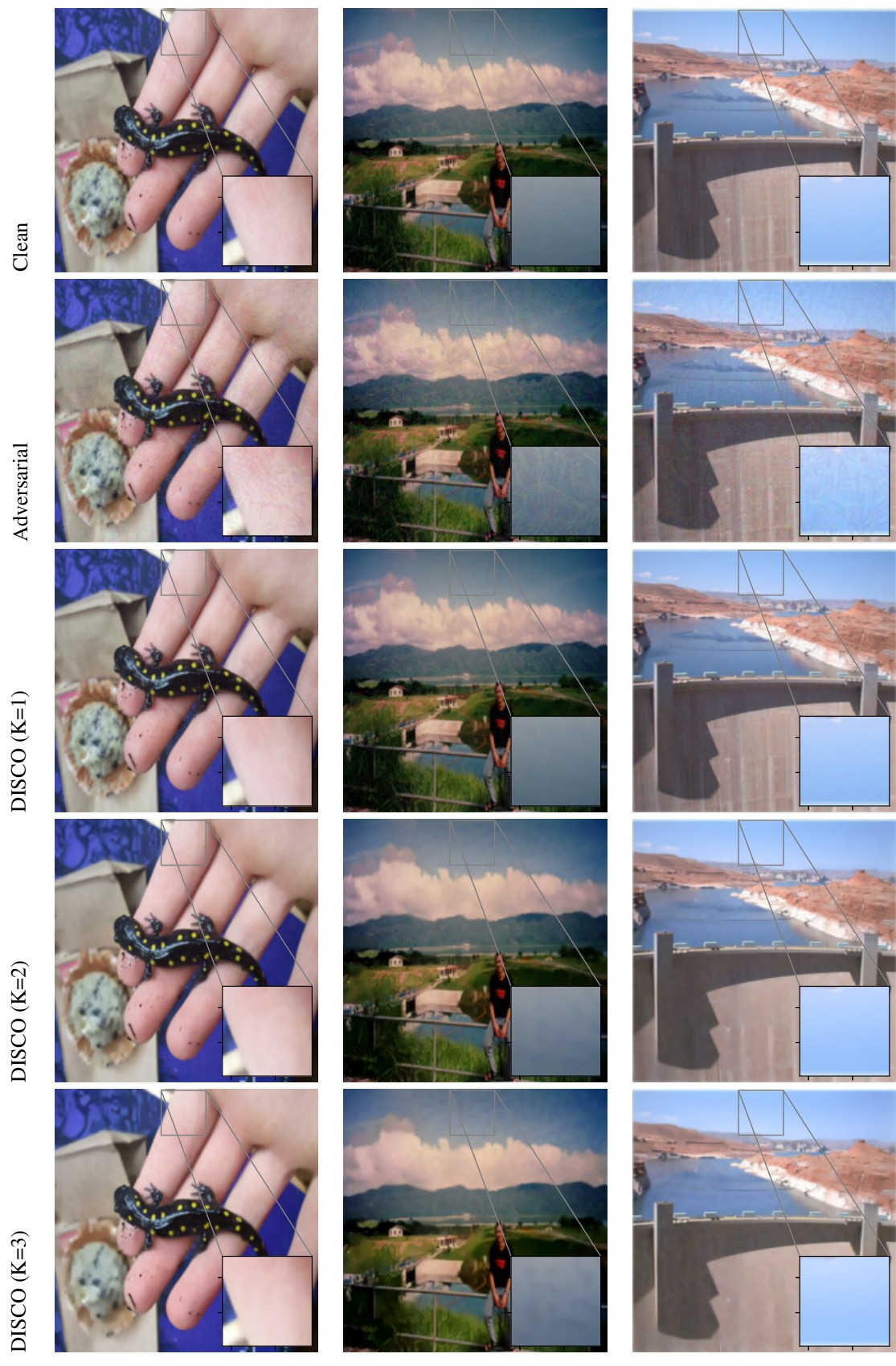

Figure F: Comparison of Clean image, Adversarial image and DISCO output from $K = 1$ to 3 under BIM attack.

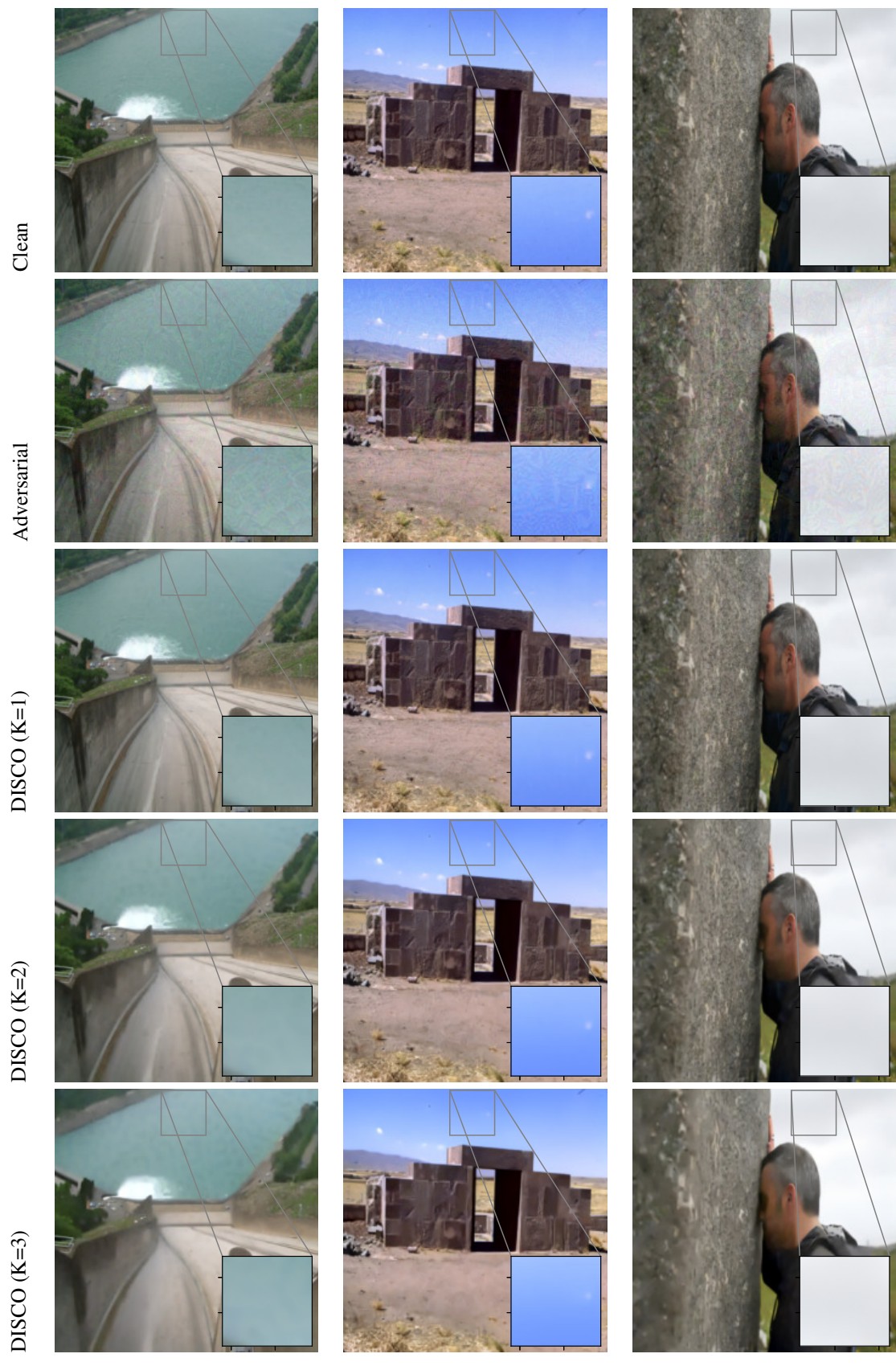

Figure G: Comparison of Clean image, Adversarial image and DISCO output from $K = 1$ to 3 under PGD attack.

Size: 128        Size: 256                    Size: 512

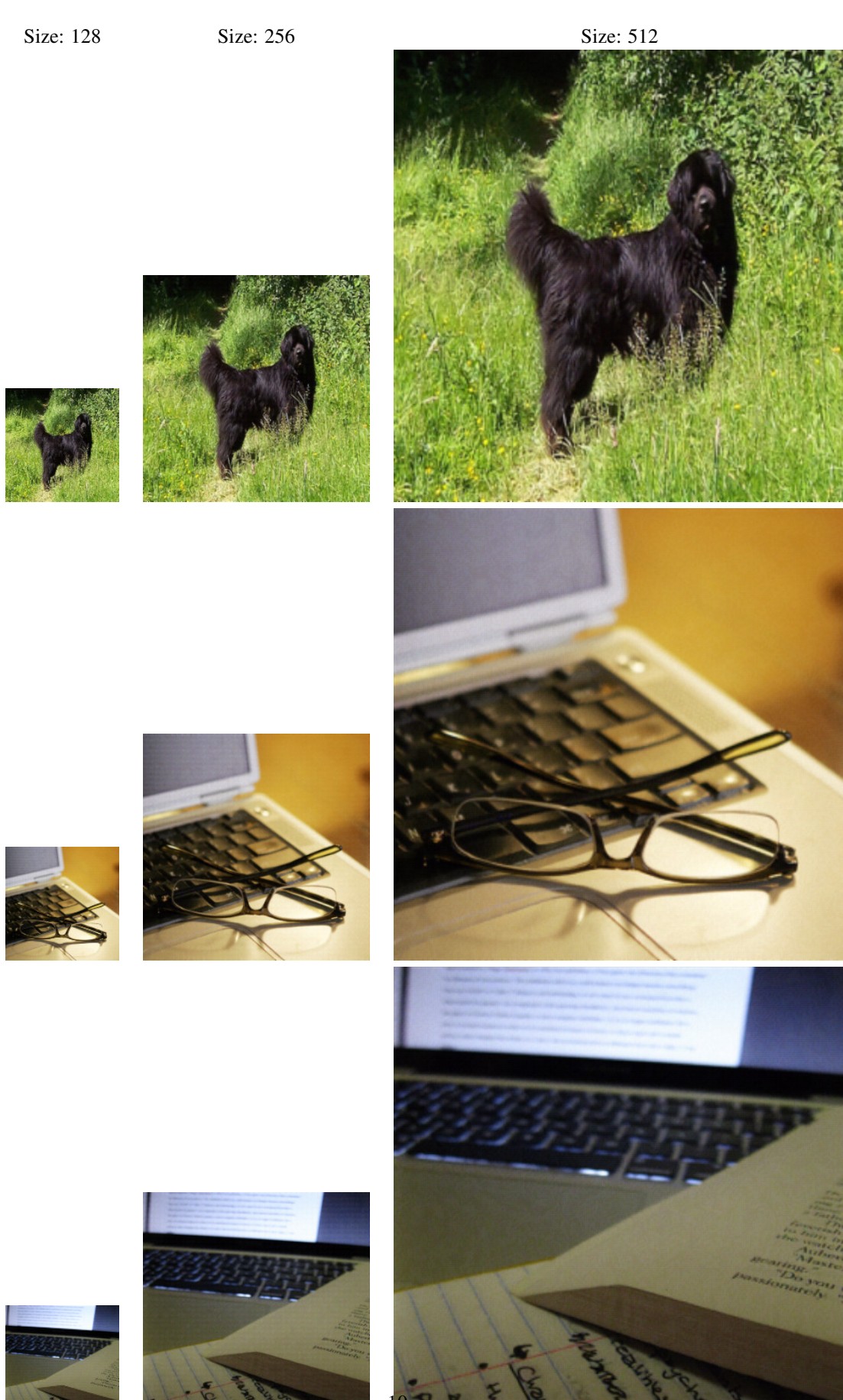

Figure H: Multiple output sizes (128, 256 and 512) of DISCO without re-training.