# OpenReview forum: "DISCO: Adversarial Defense with Local Implicit Functions"
_NeurIPS.cc/2022/Conference — NeurIPS 2022 Accept_

### Official Review · Reviewer_K7j5 · 2022-07-10

**Rating:** 4
**Confidence:** 3
**Soundness:** 3 good
**Presentation:** 2 fair
**Contribution:** 2 fair

**Summary:**

This paper provides a way, named aDversarIal defenSe with local impliCit functiOns (DISCO), to protect classifiers from being attacked by adversarial examples. DISCO is composed of two parts: an encoder and a local implicit module. For inference, DISCO takes an image (either clean or adversarially perturbed) and a query pixel location as input and outputs an RGB value that is as clean as possible. After the process, the new output image is expected to wipe out all the adversarial perturbation on it, making the classifier predicts with high accuracy. In summary, I think that DISCO is one type of denoising model that aims to be adversarially robust.

**Questions:**

1. What is the benefit of the local implicit module? Is it better than a global module only for efficient computation time?
2. What are the iterations of your PGD attack during training and testing?
3. In Table 9, why K only has 3 values for defense and attacks have 5 values?


**Limitations:**

Yes. The authors of this study addressed the limitations and potential negative social impact of their work.

**Strengths And Weaknesses:**

Strengths:

1. The experimental results are rich. Moreover, it seems that the robust accuracy of DISCO outperforms other existing methods, even for adversarially trained or transformation-based methods.
2. Transferability of DISCO: In the experiment section (Section 4), the authors assert that even though the use of datasets/classifiers/attack methods in training and testing phases are different, DISCO outperforms listed robust methods. As shown in Table 7. This particularly shows that, unlike adversarial trained, the robustness of the proposed model is independent of which attack/data be considered.


Weaknesses:

1. In lines 153-155; and lines 165-170: This should be a core part of the proposed method. However, the notations and interpretations are not clear. For example, what does "the pixel shape as input" mean? How does the local implicit module $L$ takes the three parts into consideration, to predict a clean RGB value? Moreover, I can't see any notation in Figure 5 that is used in the text. Like $E$, $L$, $\hat{f}_{i^\ast j^\ast}$, etc. So it seems like the figure does not help readers to understand the main method in this paper, but rather than making them more confused.
2. The novelty is not enough: Since the heuristic idea about adversarial removal has been developed for several years, it seems that the proposed method is mainly followed by this idea; with a slight modification.
3. Few typos listed here:

    a. In the main paper, line 144: is the generated by… → is generated by…

    b. In Supplementary, Table H, the last row: Cifar10 → Cifar100 (maybe?).

---

> ### Author Response · Authors · 2022-08-02
> **Comment to Reviewer K7j5**
>
> ### Weakness
>
> **W1:** Sorry for the confusion and thanks for the suggestion. The notation $\hat{f}_{i^*,j^*}$ denotes the concatenated feature whose location is within the kernel size s centered at $p^*$ (L165). The concatenated feature corresponds to the blue bar of dimension $C^* s^* s$ in Figure 5. The notation $E$ and $L$ are the encoder (L156-L157) and the local implicit module (L161), respectively.  More specifically, the local implicit module $L$ is a mapping of $L:\mathbb{R}^{C^* s^* s+2+2} \rightarrow \mathbb{R}^{3}$ (See figure 5), which is implemented by 4 layers MLP (each hidden layer has dimension 256) (See L162). It takes the concatenated feature (dimension $C^* s^* s$), the relative position $r=p-p^*$ (dimension 2) and the pixel shape (dimension 2) as input to predict a RGB value $v\in\mathbb{R}^{3}$. Since DISCO supports multi-resolution output, the pixel shape is the height and width of the output pixel in the normalized coordinate (See L169). Figure 5 will be revised to better match the text.
>
> **W2:** Please refer to General Comment (Novelty) for more discussion.
>
> **W3:** Thanks for the suggestions. The typos will be fixed in the final version. The last row of Table H should be Cifar100, instead of Cifar10.
>
> ### Questions
>
> **Q1:** As discussed in Section 2, implicit module has shown significant success in various field, including 2D images [13, 24] and 3D shapes [94, 68, 85, 71, 118, 52, 46, 14, 67, 77, 29, 115]. When compared to GAN based reconstruction methods in 3D, implicit module has been shown [a, 71,118, 14, 67, 77] to better reconstruct complex object details. However, even global implicit function [71,118, 14, 67, 77] is not excel in generalizing to novel object classes. Such limitation has been addressed by the introduction of local implicit function, such as [a, b, c]. To sum up, the benefit of using local implicit module for adversarial defense is multifold, including (1) better representation power of capturing local patch statistics, (2) able to transfer the defense across datasets and (3) parameter and computation efficiency (See General Comment (Efficiency)).
>
> [a] Local Deep Implicit Functions for 3D Shape
> [b] Local implicit grid representations for 3d scenes
> [c] Deep local shapes: Learning local sdf priors for detailed 3d reconstruction
>
> **Q2:** The details of PGD attack used in training and testing are identical and are applied to all the evaluated datasets (unless specified). As discussed in L49-L54 in Appendix G, we adopt the public code for the attack implementation. By default, for PGD attack, the maximum perturbation is $\epsilon=8/255$ (See L224), step size is 2/255 and the number of steps is 100. We will include more implementation details in the appendix.
>
> **Q3:** (There is no table 9 in the paper; assuming the reviewer is referring to Figure 9). As shown in Figure 9, we found $K_{def}\leq 3$ (at most 3 consecutive DISCOs) is enough to defend attacks that observes 1 to 5 DISCO stages (i.e. $K_{adv}=$ 1 to 5). The table below reports the robust accuracy with both $K_{adv}$ and $K_{def}$ from 1 to 5, which is an extension of Figure 9. Similar to the conclusion in L324-L325, the robust accuracy tends to be lower when the attacker has full knowledge of the number of DISCO cascade (i.e. $K_{adv}=K_{def}$).
>
> |  | $K_{adv}$=1 | $K_{adv}$=2 | $K_{adv}$=3 | $K_{adv}$=4 | $K_{adv}$=5 |
> | --- | --- | --- | --- | --- | --- |
> | $K_{def}$=1 | 47.2 | 55.3 | 58.9 | 62.4 | 64.2 |
> | $K_{def}$=2 | 59.6 | 52.0 | 57.5 | 57.7 | 60.4 |
> | $K_{def}$=3 | 65.4 | 59.8 | 57.2 | 58.5 | 59.2 |
> | $K_{def}$=4 | 68.6 | 60.9 | 60.0 | 57.3 | 58.5 |
> | $K_{def}$=5 | 69.4 | 64.0 | 60.3 | 58.9 | 57.7 |

---

### Official Review · Reviewer_GAdX · 2022-07-11

**Rating:** 7
**Confidence:** 4
**Soundness:** 3 good
**Presentation:** 3 good
**Contribution:** 3 good

**Summary:**

The authors propose the use of local implicit functions to undo adversarial perturbations. This is inspired by related works on image manifolds, specifically image synthesis, as well as recent progress on implicit representations for 2D/3D data. Namely, instead of processing the entire image, the authors propose to process individual pixels conditioned on local information from a small patch which is projected by an MLP to a corrected value in the spirit of implicit neural representations. Training is based on minimizing the L1 distance of the MLP output against the clean pixel RGB. An extensive set of experiments show that the proposed approach outperforms prior defenses in a number of settings, with superior transferability.

**Questions:**

1. Mainly, I disagree with the high level description of the approach as a projection on the "image" manifold while it is more of a "patch" manifold [2]. Similarly, barely outliers as used to refer to perturbed inputs is not the same thing as manipulating individual patches. This contrast is also made more apparent due to the repeated references and comparison to image synthesis and GANs. Taken together, it is not clear how global information is incorporated in the encoded features used by the implicit module, so I feel that more work is needed to explain the improved performance of the proposed model.

2. I would like to see some timing statistics. L174 ("Note that this is not computationally intensive because the encoder feature map f = E(x) is computed once and used to the predict the RGB values of all query pixel locations.") does not tell the whole story.

3. The references are somewhat excessive. While this is an appreciated effort, it is difficult for readers to trace the important ideas. I would encourage the authors to revisit the long sequences of references leaving only 2-3 in the main text and perhaps move the rest to a "see also ..." add-on remark.

4. There is quite a bit of simple language glitches. Please double-check. For example:
L128: lie a
L140: trained projecting
L144: is the generated by
L154: for each pixel locations

[2] Peyré, G. (2009). Manifold models for signals and images. Computer vision and image understanding, 113(2), 249-260.

**Limitations:**

The main limitation is the assumption of norm-bounded perturbations.

**Strengths And Weaknesses:**

Strengths:
- Disentangle the training of the defense modules from the base classifier.
- A significant saving in the number of parameters in the defense network.
- Considering the relative cost of training the defense vs attack modules, which can be magnified by repeating the projection up to K times.
- Improved robustness and transferability over known defenses.

Weaknesses:
- This is essentially an image smoothing approach which comes with the following drawbacks:
- - Each pixel-patch is passed through the MLP which is likely to slow down the inference process.
- - It assumes norm-bounded attacks leaving it susceptible to other forms of attacks as the authors mention (e.g., 1-pixel, patch) in addition to e.g., functional attacks [2].

[1] Laidlaw, C., & Feizi, S. (2019). Functional adversarial attacks. Advances in neural information processing systems, 32.

---

> ### Author Response · Authors · 2022-08-02
> **Comment to Reviewer GAdX**
>
> ### Weakness
> DISCO is specifically designed for adversarial defense, instead of image smoothing. Please refer to General Comment (Novelty).
>
> **W1 (slow inference)**: Please refer to General Comment (Efficiency) and section E in appendix for more discussion.
>
> **W1 (other attacks)**: Since we mainly evaluated DISCO against norm-bounded attacks, we did not claim DISCO's robustness or its vulnerability against other forms of attacks (e.g., 1-pixel attack, patch attack or functional adversarial attacks) (See L339-L341). This will be investigated in future work. Please refer to General Comment (Limitation) and section 5 for more discussion.
>
> ### Questions
>
> **Q1**: Sorry for the confusion and we will carefully differentiate the term "image" manifold and "patch" manifold in the final version. In fact, we do make the statement that DISCO does not project the entire image into the manifold, but only the local patch (See L61-L62 ; L72-L73; caption of Figure 2). Furthermore, we use the term barely outliers to refer to the perturbed images (L27-L29). While there are multiple defense approaches that project the barely outliers to the natural image manifold, these are usually global image modelling [66, 99, 88, 123, 95, 5, 105, 86, 53] (L33-L39), which is also adopted by many image synthesis and GANs methods. Unlike prior approaches, DISCO performs the barely outliers projection by modeling the local patch statistics and repeating the local manifold projection process over all the pixel neighborhoods. As mentioned in L61-L64 and caption of Figure 2, DISCO performs local manifold projection at each pixel neighborhood, conditional on feature vectors of the adversarial input image. The local modeling requires much smaller parameter and training dataset sizes than global modeling models and enables much more precise control of the manifold projection operation. As suggested, we will revise the text in the final version.
>
> **Q2**: Given an input image, DISCO only computes the feature map f=E(x) once and for each query pixel location, the neighbor feature at the location is extracted and used to the predict the RGB value. Note that the RGB prediction at different query pixels can be performed simultaneously in a batch. Please refer to General Comment (Efficiency) and section E in appendix for more discussion.
>
> **Q3**: Due to the excessive references and related works, we do encourage the reader to refer to related survey paper for more complete review (See L92-93). For more discussion, please refer to General Comment (Baselines) for more discussion.
>
> **Q4**: Thanks for suggestion. We will fix these language glitches accordingly in the final version.
>
> ### **Limitation**
>
> Please refer to W1 (other attacks).

---

### Official Review · Reviewer_6fhP · 2022-07-16

**Rating:** 4
**Confidence:** 3
**Soundness:** 2 fair
**Presentation:** 2 fair
**Contribution:** 2 fair

**Summary:**

This paper proposes an implicit function-based method for adversarial defense. It consumes an attacked image as input and predicts a clean image. The proposed `implicit module` uses the context information around a query pixel to reason the center pixel.

The experimental results on CIFAR10, CIFAR100 and imagenet datasets are in favor of the proposed methpd.

**Questions:**

NA

**Ethics Review Area:**

["I don’t know"]

**Limitations:**

- Novelty
- not clear the difference with denoise methods
- writing quality

**Strengths And Weaknesses:**

+. the method is clean, simple and seems effective.
+. the experimental results are in favor of the method.

- the novelty of the method is limited. there are a lot of similar methods in the field of denoising which also use similar strategy of denoise with context around. The author should clearly identify the difference between the proposed method and those denoice methods like [1, 2].
- the proposed method is tedious compared with adversarial training methods because the proposed method requires two forward passes, one for obtaining the clean image and the other for classification. While adversarial training methods requires only a single forward pass.
- The writing quality is somehow unqualified, the overall method is simple and clear, but some the writing doesn't follow a good standard. For example, in L126, you can ether remove the section description, or give more detailed one like"in this section, we xxxx, we first xxxx, then we xxxx, finally we xxx", rather than just a single sentence.


[1] Batson, Joshua, and Loic Royer. "Noise2self: Blind denoising by self-supervision." International Conference on Machine Learning. PMLR, 2019.
[2] Laine, Samuli, et al. "High-quality self-supervised deep image denoising." Advances in Neural Information Processing Systems 32 (2019).

---

> ### Author Response · Authors · 2022-08-02
> **Comment to Reviewer 6fhP**
>
> ### Weakness
>
> **W1**: DISCO is specifically designed for adversarial defense, instead of denoising. While there are many methods in the denoising literature, there is little clue that the denoise methods can be directly applied to the adversarial defense literature. Note that both [1,2] are customized for denoising purpose and do not conduct any experiment related to adversarial attack. Please refer to General Comment (Novelty) for more discussion.
>
> **W2**: Please refer to General Comment (Efficiency) and section E in appendix for more discussion.
>
> **W3**: Thanks for the suggestion. We will revise the writing as suggested in the final version.
>
> **Limitation**: Please refer to the discussion of W1 to W3.

---

### Official Review · Reviewer_QYwN · 2022-07-19

**Rating:** 7
**Confidence:** 4
**Soundness:** 3 good
**Presentation:** 3 good
**Contribution:** 3 good

**Summary:**

DISCO is a test-time defense against adversarial attack that removes perturbations from inputs by projecting them onto a local implicit representation.
The local implicit representation of an input pixel is encoded by a convolutional network then decoded by an MLP given the coordinates of the pixel and its nearest convolutional features.
This approach to image encoding and decoding follows LIIF, which proposed implicit representations for tasks like super-resolution, but not adversarial defense.
The representation is trained on paired data of clean and adversarial inputs, where the adversarial inputs are generated by a standard attack such as PGD, by sampling matching patches from the clean and adversarial inputs and minimizing the L1 distance between the decoded adversarial patches and the clean patches.
By operating on local patches and their corresponding convolutional features, DISCO is able to generalize without excessive amounts of training data or iterations, and do so with a much reduced total number of parameters compared with other defenses.
Since DISCO takes input images and makes output images, it can be composed into a cascade to leverage more computation for more robustness.
Experiments cover standard benchmarks like RobustBench, the usual datasets of CIFAR-10 and ImageNet, and a variety of defense baselines with adversarial training and older test-time transformation-based defenses.
DISCO achieves state-of-the-art robustness against the standard attacks like AutoAttack, and more distinctly shows transfer across attacks, datasets, and architectures, which distinguishes it from the dominant adversarial training approaches to defense.



**Questions:**

Questions

- Is DISCO robust to a transfer attack on the underlying classification model? That is, if AutoAttack is applied to the nominally or adversarially trained classifier without DISCO, and then these attacked inputs are presented to DISCO + the classifier, do the attacks succeed?
  This is a valid attack under a white-box threat model, and the possibility of training surrogate models. (Note: AutoAttack by default returns the original input, and not the perturbed input, so make sure to nevertheless use the perturbed inputs for this transfer attack, as near-failures may succeed when transferred.)
- How is the DISCO cascade computed? Is it simply the iterated application of the DISCO encoder and decoder on their outputs?
- Please discuss the contribution relative to LIIF [13]. It seems that DISCO should be given the empirical credit for the application of local implicit functions to defense, but much of the technical contribution was established by LIIF. (This can be fine, but clear credit attribution is a part of good scholarship.)
- Please explain how additional attack iterations are _worse_ in Figure 9. Is this not a symptom of an issue with the chosen attack? More steps should not hurt, unless the unrolling is in effect resulting in vanishing gradients, causing a kind of obfuscation that is not actually strongly measuring the defense itself.
- (Minor) What should we conclude from the sensitivity of DISCO to the number of classes for training (Table 5)? Does this suggest that DISCO requires class-comprehensive and class-balanced data if it is to be effective? Could there be way to make it more class-agnostic?

Other Feedback

- As DISCO is a test-time defense, these related strong and recent test-time defenses could be of interest:  DiffPure [F] and LINAC [G].
  To be clear this is just an FYI, and the existence of these concurrent papers publshed after the deadline at ICML'22 have no bearing on this review.
- The claim about relative complexity on lines 205-206 holds not just for adversarial training, but any defense (including test-time defenses) that take model gradients (such as self-supervised input purification, SOAP).
- Figure 2 makes an important point about local representation, but does so confusingly. Consider re-captioning the figure to highlight the point that DISCO is spatially local, and so can defend different content in the same image, and learn to defend many images by training on patches from a single image.
  Rather than pasting one image into the corner of another, why not just concatenate the images side-by-side in width? By the way, does DISCO actually fix these inputs? How is each classified?
- Figure 5 has unclear elements. What do the arrows mean in orange and red? Are they simply an ordering of the input pixels? Consider placing a box around the loss to indicate the training phase, since inference only queries the implicit module without a loss. Consider depicting the implicit module as an MLP (visualized as some stack of layers, or however) to more fully summarize the architecture of the defense.

[F] Diffusion Models for Adversarial Purification. Nie et al. ICML'22

[G] Hindering Adversarial Attacks with Implicit Neural Representations. Rusu et al. ICML'22



**Limitations:**

Empirical limitations and societal impacts are discussed. The main empirical limits are the need to try different attack families, with sparse spatial attacks suggested in this work and decision-based attacks suggested in this review, and the need to experiment on more classifiers, datasets, etc.
This amount of discussion is adequate, but it would be better still to acknowledge that the evaluated attacks (like AutoAttack) were primarily designed for train-time defenses, not test-time defenses, and so more work is likely needed to design adaptive attacks on implicit representations and other test-time methods.

**Strengths And Weaknesses:**

Strengths

- Robustness: DISCO achieves state-of-the-art robustness across datasets (CIFAR-10, ImageNet), norms (L_inf, L_2), and architectures (ResNets, Wide ResNets, different training methods).
  Furthermore, test-time defense with DISCO complements train-time defense by adversarial training, and composing the two is more robust still (Table 4).;
- Efficiency: DISCO is efficient to train and test in terms of computation, data, and parameters. As a spatially-local defense, its computation by convolution is efficient, and its generalization is improved by its smaller input dimensions (of small patches, rather than whole images).
  As a local convolutional/implicit model, DISCO only needs <0.1x the parameters of current classifiers and generative models deployed for defense, and it may be computed incrementally across pixels, so its memory usage is minimal.
  Generalization is shown by training on <1% of the training data (as patches), as opposed to the 100% normally used for adversarial training (as images), while still achieving improved robustness.
- Transferrability: DISCO is trained against PGD but evaluated on the standard RobustBench suite as well as a collection of other (generally weaker, but still diverse) attacks like BIM and FGSM. This sort of transferrability is not easily achieved for adversarial training, and such methods generally re-train the model against different attacks, which requires great computational expense.
- Attack/Defense Resource Asymmetry: DISCO requires more computation of the attacker than the defender, which may hinder the practicality of attack. This is demonstrated for gradient-based attack (Figure 10) in computation time and is also true of memory if doing full BPDA.
  However, this is not necessarily true of black-box attacks, which do not scale in the same way. That said, this point is softned by DISCO's robustness to the black-box Square attack.

Weaknesses

- The method is essentially LIIF [13], and its technical contributions w.r.t. to LIIF need to be highlighted, with credit given to LIIF for the foundation.
  While LIIF is cited in passing in the related work, this is not a sufficient or appropriate amount of acknowledgement, when Sections 3.1, 3.2, and 3.3 generally follow from it.
  One difference is that DISCO is trained for adversarial denoising, with paired clean/attacked images, but the architecture, inference, and general scheme of local implicit image functions are all due to LIIF.
  DISCO resembles LIIF all the way down into its specific implementation, with an EDSR-like architecture, and input patch size of 48x48, for example.
- DISCO harms standard accuracy by 4-5 points absolute, while competing adversarial training defenses can lose less (~2% points, for Rebuffi et al. [81] for example.)
- The related work and experiments exclude a whole wave of more recent test-time defenses some of which are also agnostic to the classifier, as DISCO is.
  Please see references [A-D] below (in chronological order of first appearance). This gap in scholarship is significant, and would be a reason for rejection if it were to go unaddressed.
- Note that many test-time defenses have claimed large boosts in robustness, but further evaluation showed the gains to be exaggerated [E].
  While the experiments in this work broadly cover different datasets, architectures, and attacks, there are nevertheless gaps.
  1. There is no decision-based attack. Square is black-box, but still depends on confidence, and RayS or Boundary have been found to succeed in cases where Square fails.
  2. There is no transfer attack from the trained classifier (without the defense) to the composition of DISCO and the classifier. Such transfer attacks, which are valid in the white-box setting, can succeed against input purification defenses like DISCO.
- (Minor) Cascade DISCO is not clearly described, though a reader may guess that it is the iterated composition of DISCO with itself, where the decoded output of one step is re-encoded as the input to the next step.

[A] Adversarial Purification with Score-based Generative Models. Yoon et al. ICML'21.

[B] Combating Adversaries with Anti-Adversaries. Alfarra et al. AAAI'22 (arXiv'21).

[C] Adversarial Attacks are Reversible with Natural Supervision. Mao et al. ICCV'21.

[D] Diffusion Models for Adversarial Purification. Nie et al. ICML'22.

[E] Evaluating the Adversarial Robustness of Adaptive Test-time Defenses. Croce et al. ICML'22 (arXived three months before the deadline, in February).

---

> ### Author Response · Authors · 2022-08-02
> **Comment to Reviewer QYwN**
>
> ### Weaknesses
>
> **W1**: DISCO first introduces implicit function for adversarial defense. While there are other ways to implement implicit function, we adopt [13] for DISCO implementation. The purpose of [13] and DISCO is entirely different, where the former/latter is designed for superresolution/defense. We modified the code of LIIF for DISCO, which consumes adversarial image (Appendix L61). We also proposed DISCO cascade for better defense. We will highlight the credit of LIIF in the main paper
>
> **W2**: The results of [81] listed in Table 1&2 are obtained with WRN70-16, while the results of no defense (first row of Table 1&2) are obtained with WRN28-10. Take Table 1 for example. When compared to [33] (Table 1 row 4) that also uses WRN28-10, DISCO beats [33] on SA (89.26 vs 87.5) and RA (85.56 vs 63.44). [81] also reports the result with WRN28-10 in Table 2 of [81], where the results of SA/RA are 89.90/62.06. Under WRN28-10, [81] harms SA by 4.88, while DISCO harms SA by 5.52. However, DISCO outperforms [81] by 23.5 (85.56 vs 62.06) on RA
>
> **W3**: While [A-D] and DISCO shares the idea of adversarial purification, DISCO is a defense that models the local patch statistics. Such property results in data and parameter efficiency, which is not shown in [A-D]. Below compares DISCO with [A-D] and DISCO beats all 4 baselines. The discussion of [A-D] will be included in the paper
>
> According to [A]’s setup, DISCO is evaluated on Cifar10 using WRN28-10 under PGD40 attack ($\epsilon=8/255$). While [A] reported SA/RA of 86.14/80.24 using its default setting, DISCO achieves 89.26/80.80. Note that DISCO is also not optimized for this experiment. DISCO also has much less param. than [A] (1.6M vs 29.7M)
>
> Under AutoAttack, [B] achieves 79.21/40.68 RA (Table 3 of [B]) on Cifar10/Cifar100 dataset, while DISCO achieves 85.56/67.93 (Table 1 & Appendix Table C). Under APgd[18] attack, [B] achieves 80.65/47.63 RA (Table 3 of [B]) on Cifar10/Cifar100 dataset, while DISCO achieves 85.79/77.33 (Appendix Table E & Table 3). DISCO beats [B] on 2 different attacks and datasets
>
> Under AutoAttack, [C] achieves 67.79/33.16 RA (Table 1&2 of [C]) on Cifar10/Cifar100 dataset, while DISCO achieves 85.56/67.93 (Table 1 & Appendix Table C)
>
> [D] also compares with SOTA defenses in RobustBench. When Cifar10 and WRN28-10 classifier is considered, [D] achieves 70.64/78.58 RA (Table 1 & 2 of [D]) under $\epsilon_\infty=8/255$ and $\epsilon_2=0.5$ respectively, while DISCO achieves 85.56/88.47 (Table 1 & Table 2). When ImageNet is considered, [D] achieves 40.93/44.39 RA (Table 3 of [D]) with ResNet50/WRN50, while DISCO achieves 68.2/69.5 (Appendix Table D)
>
> **W4 (Decision-based Attack)**: Since DISCO mainly follows the setting in RobustBench, we do not consider decision-based attack in this work. We will add this to the limitation. See general comment (Limitation)
>
> **W4 (Transfer attack)**: DISCO is robust to transfer attack. Table 4 shows that if the attacked inputs are computed using AutoAttack on classifier w/o DISCO, the attacked inputs fail when presented to DISCO+classifier
>
> **W5**: Each DISCO in the cascade shares the same weight (Fig. 3(c) & Sec. 3.4). Assume K DISCO stages. The input is passed to the 1st DISCO and the output of 1st DISCO is passed to the 2nd DISCO. The process is repeated for K times and the output of Kth DISCO is passed to the classifier
>
> ### Questions
>
> **Q1**: See W4 (Transfer attack)
>
> **Q2**: See W5
>
> **Q3**: See W1
>
> **Q4**: Fig. 9 shows the results when different numbers of DISCO stages are presented for attack and defense. When the numbers of DISCO stages are identical during attack and defense ($K_{def}=K_{adv}$), RA is lower (L324-325), because the attacker has the full knowledge of the defense. Note that the BPDA [4] attack is used in Fig. 9, which has shown to be a strong attack to circumvent defenses with obfuscation gradient
>
> **Q5**: Table 5 shows the SA and RA on ImageNet (1000 classes) when DISCO only observes partial classes (100 and 500 classes) during training. Unlike baselines that train on entire dataset, DISCO has decent results, even on unseen classes during testing. Note that each class has equal number of sample (50 per class; See data size in Table 5)
>
> ### Others
>
> **References** [A-G] will be added
>
> **Fig 2** shows that DISCO can capture local representation. In practice, DISCO purifies an image without concatenating other images in width. While the suggestion is interesting, it might cause the aliasing effect on image borders, even though the effect might not hurt the result. We will study this in the future
>
> **Fig 5**: The arrows means that the pixel location p loops over the entire image and DISCO will predict a RGB value for each pixel. The arrow direction is meaningless, because RGB prediction at different pixels can be processed simultaneously and are not dependent on each other. The implicit module is implemented by MLP (L162). Figure 5 will be revised
>
> **Limitation** will be revised

---

> > ### Comment · Reviewer_QYwN · 2022-08-08
> > **Thank you for acknowledging prior work, comparing to adaptive test-time baselines, and answering questions.**
> >
> > Thank you for the thorough response. It has adequately addressed almost all of my concerns, provided that the method and related work are amended as discussed, so that LIIF and existing test-time defenses receive adequate credit to help readers navigate this topic.
> >
> > The one remaining concern is the strength of the transfer attack evaluation. For transfer attacks, let me first highlight advice from the initial review:
> >
> > > Note: AutoAttack by default returns the original input, and not the perturbed input, so make sure to nevertheless use the perturbed inputs for this transfer attack, as near-failures may succeed when transferred.
> >
> > To unpack this further, just running the default AutoAttack on the classifier without DISCO is a weaker attack than it could be and should be. There are two issues with AutoAttack for transfer: (1) if the attack does not succeed it returns the original unperturbed input and (2) the first attack to succeed is returned. (1) is an issue because a perturbed input may still cause a misclassification when transferred, as it is applied to a different model, but the original input is obviously an easier input. (2) is an issue because the first perturbation to achieve misclassification may not be the strongest. For example, an iterate of PGD may just barely push the logit for the wrong class higher than the logit for the right class, and at this point AutoAttack will terminate. A stronger transfer attack would keep iterating PGD up to some threshold number of steps to try and make the loss even higher. While (2) is a good efficiency trick when used without transfer, it can result in overestimates of robustness when used with transfer.
> >
> > The bottom line is that _running default AutoAttack is not a sufficient transfer attack_. The four attacks it includes are good choices, but these cannot be run with the default library configuration, or else the transferred attacks will not be as strong as they could be. I encourage the authors or future readers to double-check the transfer results to achieve potentially more accurate estimates of robustness.
> >
> > At this point I am encouraged to maintain my rating (7/Accept). Although DISCO is limited in its technical novelty, it is empirically novel, and it is highly informative to the community to evaluate robustness across so many models, datasets, and attacks while emphasizing transferrability across classifiers and attack types. If DISCO's robustness holds up to further evaluation, the kind of test-time defense proposed here could be a revealing counterpoint to the mainstream of adversarial training.

---

### Official Review · Reviewer_wEq1 · 2022-07-28

**Rating:** 6
**Confidence:** 4
**Soundness:** 3 good
**Presentation:** 3 good
**Contribution:** 3 good

**Summary:**

The authors propose DISCO to remove adversarial perturbations by localized manifold projections. They aim to output the clean RGB value for an adversarial image and a pixel location. Their method is built upon the assumption that the manifold projection required for adversarial defense is conditioned on the synthesis of a natural image given the perturbed one which can be defined as a function of local image patches instead of a the whole image.

**Questions:**

1-- Can the performance be compared to these works?
a) Xu et al. "Adversarial defense via local flatness regularization"
b) Li et al. "Semi-supervised robust training with generalized perturbed neighborhood"
c) Bai et al. "Clustering effect of adversarial robust models"

**Limitations:**

The authors have included an important limitation for their work and it an interesting problem to study.

**Strengths And Weaknesses:**

Strengths:
1-- The paper is well-written,
2-- The studied problem is an interesting problem,
3-- Extensive experimental analysis.

Weakness:
1-- Lack of sufficient novelty,
2-- Lack of theoretical analysis

---

> ### Author Response · Authors · 2022-08-02
> **Comment to Reviewer wEq1**
>
> ### Weakness
>
> **W1**: Please refer to General Comments (Novelty).
>
> **W2**: Unlike some of the transformation based baselines [3, 5, 66, 88, 95, 99, 123] that project the adversarial image into the natural image manifold by modeling global statistics, our work introduces a novel perspective, based on local reconstruction by leveraging the ability of local implicit functions to perform sophisticated modeling of image statistics. Our results, namely the showing of significantly better transfer across datasets and much greater parameter efficiency, suggests that the combination of a less ambitious task (modeling of local rather than global statistics) and a more powerful local modeling (implicit function instead of GAN or AutoEncoder) is a better trade-off than those of previous approaches. However, and although local implicit functions have been quite successful in many domains, including 2D image super-resolution [13] and 3D reconstruction [94, 68, 85, 71, 118, 52, 46, 14, 67, 77, 29, 115], it is difficult to prove theoretically why the implicit function is a better model for capturing local statistics than GANs or AutoEncoders. We believe that this is a research problem on its own, which could benefit multiple domains. Our results certainly show advantages for local modeling with implicit functions, and will likely inspire theoretical work in this question. While we intend to investigate these issues, we leave the theoretical analysis for future work. We believe the paper already demonstrates the power of implicit functions as a solution to the adversarial defense problem. Note that this is the first paper to introduce implicit functions to the adversarial defense problem.
>
> ### Questions
>
> **Q1**: Please refer to General Comments (Baselines) for more discussion. While all [a-c] are related to adversarial defense, their settings are different to that considered in RobustBench[16], which proposed a fair benchmark for evaluation across defenses. More specifically, [a] only considers a single dataset Mnist in their work and its github clearly reveals its failure on Cifar10 dataset ([https://github.com/Uooga/Local-Flatness-Regularization](https://github.com/Uooga/Local-Flatness-Regularization)). [b] and [c] reports CIFAR10 robust accuracy of 47.24 (Table II of [b]) and 52.54 (Table 3 of [c]) under PGD20 attack with step size 0.003. To compare with [b,c], we apply DISCO on standard ResNet18 and achieve robust accuracy of 67.50 on CIFAR10, which outperforms [b,c] by more than 14 points. References [a-c] will be added.

---

### Author Response · Authors · 2022-08-02
**General Comment**

We thank the reviewers for their thoughtful comments. Major issues are addressed here; minor suggestions will be fixed. General comments are covered here; individual questions are addressed below. SA/RA denotes standard/robust accuracy.

1. **Novelty**:  At a high level, DISCO resembles baselines that perform adversarial removal or denoising (L107-116). While the idea of adversarial removal has been used and implemented by prior works [99, 88, 123, 95, 5], these methods model the image globally with GANs or conditional models of pixel statistics (L32-41 ; L110-116). Prior works [25,119] inspired by the denoising literature have poor performance and there is little evidence that directly applying denoising methods to adversarial defense can succeed. The restriction of the manifold modeling to small patch is a critical difference between DISCO and prior defenses based on image manifold modeling  (L61-62). The use of implicit function to implement the manifold projection also enables DISCO to model the conditional local image statistics more accurately, which is the key difficulty of prior works. Due to these 2 properties, DISCO outperforms prior works by a large margin (Table 1, Table 2 and Fig. 6), can produce outputs of various sizes (L116) and can transfer the defense across datasets (L292-302). Overall, while DISCO shares the spirit of prior methods, we believe the perspective of modeling local statistics and the introduction of the local implicit function for adversarial defense are important contributions.
2. **Baselines**: While we appreciate the additional baselines suggested by the reviewers, we would like to emphasize that DISCO is already compared and outperforms 120+ models on the RobustBench[16]. As mentioned by [16], there are more than 3000 papers in the adversarial literature and many of them are evaluated under different criteria. So, fair comparisons are not always easy, which led to the introduction of [16]. Hence, we mostly compare DISCO with baselines on [16] for fair comparisons. Because there are so many results, we intentionally place the tables of numerical results in the appendix and only include the corresponding plots (Fig. 6) in the main paper. This is to help readers focus on the important ideas. As suggested, we will further shorten the references list and keep only the essential ones.
3. **Efficiency**: As discussed in Sec. 3.4 (L189-202), DISCO is parameter efficient (1.6M) compared to SOTA classifiers (for example ResNet101 has 44.5M; See Fig. 4), because DISCO only operates on local patches instead of entire image. Given this efficiency, the defense complexity can be analyzed with respect to (a) training and (b) testing phase. Consider the ImageNet, for example.

    (a) For training, DISCO only requires 0.5% training data (L195), while the adversarial training methods require the entire training set. In addition, unlike adversarial training methods that compute adversarial examples on-the-fly, we precompute the adversarial images in the data preparation stage (See Fig. 3(a)), which expedites the training process. Finally, since DISCO is classifier agnostic (See L287-291), no forward pass through the classifier is required during training. All together, these properties make DISCO significantly efficient to train.

    (b) For testing, the adversarial examples have to be passed through DISCO + classifier, resulting in a memory cost of $O(N_c + N_d)$ (See L203-L207). However, it is usually the case that $N_c$ > $N_d$ (See Fig. 4), making the complexity of the classifier larger than that of DISCO. In addition, Table 4 and Table H of the appendix show that DISCO + robust classifier further improves RA. For clarity, we compare the FPS of SOTA method [81] and DISCO+[81] (See Table H), on our machine (L218). The former achieves 33.7 FPS and the latter 29.3 FPS. While adding DISCO lowers the FPS by 4.4, it also increases the RA by 4.13 points (66.58 vs 70.71; See Table H). On the other hand, when compared to STL [99] (See Table J and section E in appendix), DISCO is 5.9x faster. To sum up, while adding DISCO has a slight increase in computing cost, this cost is minor and enables a large increase in RA. We believe that the latter outweighs the former and DISCO is superior when all factors are considered.

4. **Limitations**: DISCO is mainly evaluated under norm-bounded attacks and we leave the investigation of other type of attacks for future work (L339-343). While it would be ideal to develop a universal defense, robust to all types of attacks (patch attack, decision based attack, functional adversarial attack, etc.), most defenses are only evaluated on certain types of attacks. For example, many baselines in RobustBench [25,26,32,33,81,87,99,110,116,119,116] are evaluated only on norm-bounded attacks, without even discussing other types of attacks. We hope that our disclosure of DISCO’s limitation will not be a reason to penalize it.

---

> ### Author Response · Authors · 2022-08-09
> **Author Rebuttal Acknowledgement. Thank you**
>
> Dear Reviewers,
> We appreciate your efforts in reviewing our paper. We have addressed your questions in detail. As the deadline is approaching, would you please check our response and acknowledge our rebuttal?
> Thank you so much.
> Best regards,
> Authors

---

### Meta-Review · Area_Chair_UdYp · 2022-08-25

**Recommendation:** Accept
**Confidence:** Certain

**Metareview:**

In this paper, DISCO, a test-time defense against adversarial attack, is proposed based on prior concents of adversarial denoising, manifold modeling, and implicit function.
The authors show promising efficiency and experimental results in DISCO.

However, a large concern raised by some reviewers is the limitied novelty but the authors claimed that the perspective of modeling local statistics and the introduction of the local implicit function for adversarial defense are important contributions.
Another limitation is that some reviewers concern robustness evaluation on norm-bounded attacks only, but the authors claim that many baselines in RobustBench [25,26,32,33,81,87,99,110,116,119,116] are evaluated only on norm-bounded attacks.

Since most reviewers are satisfied with authors' responses, this work is suggested to be accepted but the AC hopes the authors continue to clarify the limitations and consider taking recent publications into consideration to further revise the paper.

**Award:**

No

---

### Decision · Program_Chairs · 2022-09-14

Accept